# Synthesis of Bio-Inspired 1,3-Diarylpropene Derivatives via Heck Cross-Coupling and Cytotoxic Evaluation on Breast Cancer Cells

**DOI:** 10.3390/molecules27175373

**Published:** 2022-08-23

**Authors:** Aik Sian Tan, Jaymeer Singh, Nurul Syafiqah Rezali, Musthahimah Muhamad, Nik Nur Syazni Nik Mohamed Kamal, Yvan Six, Mohamad Nurul Azmi

**Affiliations:** 1School of Chemical Sciences, Universiti Sains Malaysia, Gelugor 11800, Penang, Malaysia; 2Chemical Sciences Programme, School of Distance Education, Universiti Sains Malaysia, Gelugor 11800, Penang, Malaysia; 3Integrative Medicine Cluster, Advanced Medical and Dental Institute, Universiti Sains Malaysia, Kepala Batas 13200, Penang, Malaysia; 4Laboratoire de Synthèse Organique (LSO), École Polytechnique, CNRS, Institut Polytechnique de Paris, CEDEX, 91128 Palaiseau, France

**Keywords:** Heck cross-coupling, 1,3-diarylpropene, β-hydride elimination, cytotoxic effects, breast cancer cells

## Abstract

The Heck cross-coupling reaction is a well-established chemical tool for the synthesis of unsaturated compounds by formation of a new C-C bond. In this study, 1,3-diarylpropene derivatives, designed as structural analogues of stilbenoids and dihydrostilbenoids, were synthesised by the palladium-catalysed reactions of 2-amidoiodobenzene derivatives with either estragole or eugenol. The products were obtained with high (*E*) stereoselectivity but as two regioisomers. The ratios of isomers were found to be dependent on the nature of the allylbenzene partner and were rationalised by electronic effects exercising a determining influence in the β-hydride elimination step. In addition, the cytotoxic effects of all the Heck reaction products were evaluated against MCF-7 and MDA-MB-231 human breast cancer cells, with unpromising results. Among all, compound **7d** exhibited weak cytotoxic activity towards MCF-7 cell lines with IC_50_ values of 47.92 µM in comparison with tamoxifen and was considered to have general toxicity (SI value < 2).

## 1. Introduction

The Heck reaction has perpetuated its significance in synthetic organic chemistry for over half a century. Indeed, the introduction of this carbon-carbon bond-making cross-coupling reaction by Heck and Mizoroki has revolutionised synthetic methodology, thanks to its simplicity and effectiveness [1,2,3,4,5]. Since the first detailed mechanism was proposed by Heck and Nolley in 1972 [2], the elementary steps, reaction parameters and catalytic systems have counted among the major aspects attracting the interest of many researchers [4,6]. One crucial part of the Heck catalytic cycle, that sometimes receives little attention by synthetic chemists, is the β-hydride elimination elementary step. This process plays a key role in enantioselective versions of the reaction, for which it must proceed with suitable regioselectivity [7,8,9,10]. Control is also essential in so-called reductive Heck reactions for which, conversely, β-hydride elimination is to be avoided [11]. In relation to this context, the synthesis of 1,3-diarylpropene derivatives by the Heck reaction is especially worthy of study. Indeed, when the two aromatic groups differ, two possible regioisomers can be produced (Figure 1).

Moreover, such compounds can be viewed as higher homologues of 1,2-diphenylethylene derivatives (stilbenes), as well as of dihydrostilbenes. In this respect, it is worth pointing out that hydroxylated stilbenes (stilbenoids) and hydroxylated dihydrostilbenes are of special biological importance. Indeed, stilbenoids are produced by various plants in response to stressors such as mechanical injury, e.g., resulting from attack by herbivores, or infection by pathogenic agents [12,13,14]. Most of these phytoalexin molecules are derived from *trans*-resveratrol, the presence of which in grape vines has been thought to be involved in the “French paradox” [15]. (*E*)-Resveratrol and other naturally occurring (*E*)-stilbene derivatives, such as piceatannol, oxyresveratrol or isorhapontigenin, have attracted much attention and have been the subject of extensive studies. Various interesting pharmacological properties have been disclosed, including antioxidant, anticancer, anti-inflammatory, cardioprotective, antifungal and antibacterial activities [14,15,16,17,18,19]. Interestingly, dihydrostilbenoids have been reported to exhibit a similar range of biological effects [20]. Some years ago, we studied the preparation of stilbenoid structural analogues having an amido substituent at the *ortho* position of one of the two aromatic parts, using the Heck reaction [21,22]. Some of these molecules were found to possess significant anticancer properties against HT-29, P388, DU-145, MCF-7 and BxPC-3 cancer cell lines, or chemopreventive action [21,22].

In our continuing efforts directed towards the discovery of novel biologically active derivatives, we decided to investigate the application of the Heck reaction to the synthesis of new compounds having a 1,3-diarylpropene skeleton, thus moving from a C6-C2-C6 to a C6-C3-C6 structure, but retaining the pharmacophores, i.e., an amido substituent on one of the phenyl rings and oxygen-based substituents on the other (Figure 2). It is noteworthy that these new target molecules can be viewed not only as structural analogues of stilbenoids and dihydrostilbenoids, but, in fact, as hybrid compounds having characteristics of both types of molecules.

## 2. Results and Discussion

### 2.1. Synthesis of Amido-Substituted 1,3-Diarylpropene Derivatives

The synthesis of the targeted *o*-amido-substituted 1,3-diarylpropene products is illustrated in Figure 2. First, the amido-substituted iodobenzene precursors **3a**–**j** were prepared by *N*-acylation of 2-iodoaniline **1** with the respective acyl chlorides **2a**–**j**. These compounds were then cross-coupled with estragole **4** or eugenol **5** using the Heck reaction, in the presence of palladium(II) acetate as a catalyst, affording the expected *o*-amido-substituted 1,3-diarylpropene products **6a–j**–**9a–j** in 36–91% yields (Table 1). In each case, the formation of both possible regioisomers, **6**/**7** or **8**/**9**, was observed [23,24,25].

### 2.2. Characterisation of the 1,3-Diarylpropene Heck Products

Compounds **6a–j**–**9a–j** were characterised by FT-IR, 1D- and 2D-NMR spectroscopies as well as an HRMS analysis. Compounds **8b**,**9b** were taken as an example to describe the obtained data. The FT-IR spectrum of the **8b**/**9b** mixture exhibited characteristic absorptions at 3398 (broad, w), 3291 (broad, m), 2970–2873 (w), 1654 (s), 1509 (s) and 1371 (m) cm^−1^, corresponding to O-H stretching, N-H stretching, Csp^3^-H stretching, saturated amide C=O stretching, C=C aromatic stretching and CH_3_ bending, respectively. An out-of-plane C-H bending vibration at 753 cm^−1^ (s) indicated the presence of *ortho-*substituted aromatic rings in compound **8b**/**9b**. On the other hand, a CH_2_ bending vibration was observed at 1446 cm^−1^ (m), indicating the presence of a methylene group (CH_2_). The *para*-substituted aromatic ring displayed out-of-plane C-H bending vibration bands at 812 (w) and 848 (m) cm^−1^.

Typically, the comprehensive description of all the ^1^H and ^13^C NMR signals of mixtures of isomers such as **8b**/**9b** was not straightforward. However, it could be achieved by first carefully assigning the signals of the few Heck products for which we had managed to obtain samples of pure isomers, including **9b**. This was done on the basis of 1D and 2D NMR experiments: COSY, DEPT135, HSQC and HMBC, as well as by comparison with reported NMR data of simpler compounds sharing structural subunits with our molecules, in particular 1-[(*E*)-cinnamyl]-4-methoxybenzene [23], 1-methoxy-4-[(*E*)-3-phenylprop-1-enyl]benzene [24], 4-[(*E*)-cinnamyl]-2-methoxyphenol [25], *N*-[2-[(*E*)-hex-1-enyl]phenyl]acetamide [26], *N*-(2-allylphenyl)acetamide [27] and *N*-[2-[(*E*)-cinnamyl]phenyl]prop-2-enamide [28]. Once this operation was completed, the corresponding descriptions were used as reference data for the remaining analyses.

The ^1^H NMR spectrum of the **8b**/**9b** mixture exhibited important characteristic signals within the range δ_H_ 1.20–7.95 ppm. A downfield broad singlet for the amide proton (NH) of the major isomer **9b** was observed at *δ*_H_ 7.35 ppm. The seven aromatic protons were distributed in two regions: 7.06−7.95 ppm and 6.72−6.85 ppm. The first area was assigned to the protons attached to the nitrogen-substituted aromatic ring. The doublet of the proton at the *ortho* position relative to the nitrogen atom was characteristic, with a particularly high chemical shift value: *δ*_H_ 7.95 ppm (*J* 8.0 Hz) for **9b** and *δ*_H_ 7.84 ppm (*J* 7.6 Hz) for **8b**. The somewhat lower value for the latter was a general observation in all the **6**/**7** and **8**/**9** series. The aromatic signals lying in the higher field region were assigned to the methoxyphenol part. They appeared as two sets of overlapping signals: a singlet, accounting for the proton located *ortho* to the methoxyl substituent; and an AB system for the other two protons, with a *J*_AB_ coupling constant of 9.0 Hz. These signals were observed at a markedly lower field in the case of **9b** (*δ*_H_ 6.81−6.87 ppm) than in the case of **8b** (*δ*_H_ 6.72−6.74 ppm), which was a general trend for all **8**/**9** mixtures.

For each isomer and as expected, the propene unit bridging the two aromatic rings appeared as a doublet (one proton, *J* 16.0 Hz) for the vinylic CH directly attached to one of the aromatic rings (*δ*_H_ 6.38 ppm for **9b** and 6.42 ppm for **8b**), a doublet of triplets (one proton, *J* 16.0 and 6.0 Hz) for the central vinylic CH (*δ*_H_ 6.16 ppm for **9b** and 6.27 ppm for **8b**) and a doublet (two protons, *J* 6.0 Hz) for the allylic protons, in the aliphatic region (*δ*_H_ 3.53 ppm for **9b** and 3.50 ppm for **8b**). The high ^3^*J* coupling constant of 16.0 Hz confirmed the *E* configuration of the CH=CH double bond. The hydroxyl (OH) and methoxyl (OCH_3_) protons appeared as two singlets at *δ*_H_ 5.73 and 3.87 ppm, respectively, in the case of **9b**. The OH proton of **8b** came out at a somewhat higher field (*δ*_H_ 5.61 ppm), which was again a general trend in the **8**/**9** series.

The isopropyl group of **9b** was characterised by an upfield doublet at *δ*_H_ 1.21 ppm (*J* 7.0 Hz), accounting for the magnetically equivalent six protons of the methyl groups. A septet (*J* 7.0 Hz) at *δ*_H_ 2.47 ppm was assigned to the neighbouring CH. It is worth mentioning that in the ^1^H NMR spectrum of the **8b**/**9b** mixture, as well as in several other cases, traces of other products could be detected. Their isolation and characterisation were not attempted, but by analogy with related results reported in the literature [24,25,26], these were likely to be minor by-products of the Heck reaction, namely *Z* isomers of **8b** and **9b** and/or the vinylidene product resulting from alternative regioselectivity of the alkene 1,2-insertion step.

The ^13^C NMR spectrum of **8b**/**9b** exhibited important characteristic signals within the range *δ*_c_ 19.6–175.1 ppm. A total of two sets of 19 carbon signals were observed, with different intensities, with the highest peaks accounting for the major isomer **9b**. The presence of the carbonyl carbon was evidenced by a downfield signal at *δ*_c_ 175.1 ppm. The signals of the twelve aromatic ring carbon nuclei of **8b**/**9b** were observed in the range *δ*_c_ 107.9−146.7 ppm. The peak of the CH located at the *ortho* position relative to the methoxyl substituent was especially characteristic, with a comparatively low chemical shift value due to the +*M* electron-donating effect of the substituents of this ring (*δ*_c_ 107.9 ppm for **9b** and 111.2 ppm for **8b**). The signals of the propene subunit also exhibited marked differences depending on the isomers. The CH_2_ group was clearly identifiable, with a chemical shift around 39 ppm for all compounds **8a–j** (*δ*_c_ 39.2 ppm for **8b**) and a value around 36 ppm for **9a–j** (*δ*_c_ 36.1 ppm for **9b**). The chemical shifts of the vinylic carbon atoms were also good markers. For instance, the central vinylic CH appeared at around 134 ppm in the **8** series (*δ*_c_ 134.2 ppm for **8b**) and at around 125 ppm in the **9** series (*δ*_c_ 125.1 ppm for **9b**). The methoxyl substituent (OMe) gave a signal at *δ*_c_ 55.9 ppm for **9b** and 56.0 ppm for **8b**; the isopropyl group was identified by two peaks at *δ*_c_ 36.7 (CH) and 19.6 ppm (Me) for both isomers.

### 2.3. Mechanistic Interpretation of the Results of the Heck Cross-Coupling Reactions

An examination of the literature reveals several examples of related Heck reactions giving mixtures of isomers [29,30,31,32,33,34,35,36]. In several cases, the ratios of isomers were not given and generally, the origin of the selectivities observed was not discussed. In a few instances, the observation of two main products having the same atom connections was interpreted as the formation of *E* and *Z* isomers [31,33], whereas those compounds are perhaps more likely to be regioisomers, as in the present work.

The mechanism of the Heck reaction and the exact nature of the intermediate palladium complexes involved have been extensively studied and discussed [3,4,5,6]. A simplified view is displayed in Figure 3. A pre-activation step generated palladium(0), which then underwent the oxidative addition of the aryl iodide substrate **3**. This produced the intermediate complex **10**, where the palladium atom was attached to the *ortho* amido-substituted aryl group. Coordination of the allylbenzene reactant **4** or **5**, followed by *syn* 1,2-insertion of the C=C bond, gave rise to the alkylpalladium species **11**. The next step was a *syn* β-hydride elimination which, after de-coordination, delivered the product: **6** or **7** from **4** or **8** or **9** from **5**, along with a palladium(II) hydride species. Finally, the palladium(0) catalyst was regenerated in a reductive elimination process.

The crucial elementary step with respect to the stereo- and regioselectivity of the formation of the products was the *syn* β-hydride elimination event taking place from intermediate **11**. Indeed, the outcome was fully determined by which of the four neighbouring hydrogen atoms that are located at the β position relative to the metal centre participates in this process. In our reactions, little regioselectivity was observed when 1-allyl-4-methoxybenzene (estragole) **4** was employed, whereas with 1-allyl-3-methoxy-4-hydroxybenzene (eugenol) **5**, selectivity in favour of product **9** was substantial, ranging from 62:38 to 81:19. In all cases, the (*E*) isomers were produced with high selectivity. The latter stereoselectivity is commonly observed in Heck reactions and it is generally considered that this is the result of kinetic control governed by the Curtin–Hammett principle [3,4,5].

Similarly, it appears that in the present examples, the observed regioselectivities cannot be explained by thermodynamic control. Products **7** and **9** were expected to be stabilised by conjugation of the C=C alkene fragment with the oxygen-substituted aromatic group, whereas in **6** and in **8**, the presence of the *ortho* amido substituent forced the conformation of the neighbouring alkene out of the plane of the aromatic ring, thereby compromising a similar stabilising conjugation effect. Semi-empirical PM7 calculations, performed with R^1^ = Me or *i*Pr, support this analysis: **7a**, **7b**, **9a** and **9b** were estimated to be more stable than their respective regioisomers **6a**, **6b**, **8a** and **8b** by 0.87 to 1.83 kcal.mol^−1^. Using estragole **4**, thermodynamic control would thus result in significant **6**/**7** selectivity in favour of **7**, in contrast to what was experimentally observed. More importantly, the energy difference between **8a** and **9a** was found to be lower than between **6a** and **7a**: 0.87 and 1.39 kcal.mol^−1^, respectively. The same situation was calculated in the case of **8b**/**9b** vs. **6b**/**7b**, with respective energy differences of 1.37 and 1.83 kcal.mol^−1^. Pure thermodynamic control would thus lead to lower selectivities with eugenol **5** than with estragole **4**, which is the opposite of what is actually observed. Therefore, regioselectivity is most plausibly under kinetic control. In the reaction of intermediate **11**, four possible transition states can be drawn, among which two, **12** and **13**, account for the formation of the observed major (*E*) isomers (Figure 4).

The higher selectivity observed using eugenol **5** indicates that the transition state **13** became comparatively more energetically accessible than **12** when the oxygen-substituted aromatic ring (displayed in blue) became more electron-rich. This was consistent with a partial positive charge being developed at the carbon atom of the C−H bond that was being broken in the process. This positive charge was best accommodated by the most electron-rich adjacent aromatic ring. This provides new support for a similar explanation that had been proposed by the group of Sigman to explain the regioselectivities of Heck cross-coupling reactions of allylbenzene with aryldiazonium salts having various electronic properties [34].

### 2.4. Cytotoxicity Activity

The MTT assay was used to evaluate the toxicity of *N*-(2-cinnamylphenyl)amides on the viability of two types of breast cancer cell lines, i.e., MCF-7 (with a receptor) and MDA-MB-231 (without a receptor). In this study, a normal breast cell line (MCF-10A) was used. MCF-7 is an ER-positive breast cancer cell, whereas MDA-MB-231 is an ER-negative breast cancer cell. These cancer cells have receptor sites that bind to oestrogen, promoting their growth and spread. Tamoxifen is a first-line anti-oestrogen therapy that is used to prevent the production or action of oestrogen.

For the cytotoxicity assays, only pure isomers were submitted for evaluations. The cell lines were treated with different concentrations of the synthesized compounds, i.e., between 10 and 100 µM and incubated for 24−72 h. Table 2 displays the IC_50_ values, as well as the SI indexes. At 72 h, compound **6i** exhibited weak cytotoxic activity against both MCF7 and MDA-MB-231 cells, with IC_50_ values of 91.78 µM and 73.79 µM, respectively. In addition, compounds **6d**, **7d** and **7e** exhibited weak activity towards both MCF-7 and MCF-10A at 72 h. The IC_50_ values recorded were 65.83 µM, 47.92 µM, 88.46 µM and 92.89 µM, 71.19 µM and 69.18 µM, respectively. They were, however, inactive against MDA-MB-231 cells. All other compounds (i.e., isolated pure isomers) tested were inactive for cytotoxicity against all cell lines.

Tamoxifen, on the other hand, demonstrated strong activity against MCF-7, MDA-MB-231 and MCF-10A, with IC_50_ values at 72 h of 15.78 µM, 14.53 µM and 4.07 µM, respectively. The SI value was obtained by dividing the IC_50_ value for normal cell lines by the IC_50_ for cancerous cell lines. All compounds with SI values less than 2 were considered to have general toxicity, suggesting that they can cause cytotoxicity in normal cells as well [37].

## 3. Materials and Methods

### 3.1. General

All spectral data were procured using the following instruments: Fourier-transform Infrared (FTIR) spectra were obtained using a PerkinElmer 2000 FTIR Spectrum spectrometer (PerkinElmer, Waltham, MA, USA). Nuclear magnetic resonance (NMR) spectra were recorded on a Bruker AVN 500 MHz spectrometer (Bruker Bioscience, Billerica, MA, USA). Data were analysed with the Top Spin 3.6.1 software package (Bruker Bioscience, Billerica, MA, USA). Chemical shifts were internally calibrated using the residual CHCl_3_ solvent peak in CDCl_3_ (^1^H, δ 7.26), the CDCl_3_ solvent signal (^13^C, δ 77.0) or the tetramethylsilane (TMS) signal (^1^H or ^13^C, δ 0.00 ppm). Tetrahydrofuran (THF, QRëC, Grade AR) was freshly distilled from sodium benzophenone ketyl. Dimethylformamide (DMF, QRëC, Grade AR) was dried over 4 Å molecular sieves prior to use. All other commercially available chemicals and materials were used without further purification. Column chromatography was performed using Merck silica gel (0.040–0.063 mm) (Merck & Co., Kenilworth, NJ, USA). Thin Layer Chromatography (TLC) was completed with silica gel-coated aluminium sheets (silica gel 60 F_254_).

The procedure of intermediates and final compounds synthesis and their NMR; ATR-IR and HRMS spectra of the compounds are presented in Appendix A.

### 3.2. Synthesis Procedures

#### 3.2.1. General Procedure for the Preparation of the Ortho-Amido-Substituted Iodobenzene Precursors

A solution of 2-iodoaniline **1** (1.00 equiv.) and Et_3_N (1.00 equiv.) in dry THF (1.6–2.0 mL per mmol of **1**) was cooled to 0–5 °C. The requisite acyl chloride **2** (1.00 equiv.) was then added dropwise, with stirring. The cold bath was removed, and the mixture was stirred vigorously overnight at room temperature. The white precipitate of Et_3_N.HCl was filtered off and rinsed with THF (3 × 5 mL). All the organic fractions were combined and concentrated under reduced pressure to afford the crude amide product **3**, which was then purified by recrystallisation from *n-*hexane/chloroform. Full details and characterisation data for compounds **3a**–**3j** are attached in the Appendix A.

#### 3.2.2. General Procedure for Preparing the Heck Cross-Coupling Reaction Products

In a dry three-necked flask equipped with a thermometer and a condenser, a solution of *N*-(2-iodophenyl) amide **3** (1.00 equiv.) in dry DMF (4.0 mL per mmol of substrate **3**) was heated up to 120 °C and stirred for 20 min under nitrogen. Palladium(II) acetate (1.00% equiv.), triethylamine (3.50–5.00 equiv.) and the requisite allylbenzene derivative **4** or **5** (1.60 equiv.) were then added successively. The reaction mixture was stirred at 120 °C until the TLC analysis indicated complete consumption of **3** (3–6 h). After cooling, the saturated NH_4_Cl aqueous solution (6.0 mL per mmol of starting **3**) was added and the mixture was extracted with EtOAc (three times, with portions of 2.0 mL per mmol of starting **3**). The combined organic fractions were washed with H_2_O (6.0 mL per mmol of starting **3**), dried over anhydrous sodium sulphate, filtered and concentrated under reduced pressure. The ratio of product isomers was determined by ^1^H NMR spectroscopy of the crude product. Purification by gradient elution column chromatography (*n-*hexane/ethyl acetate, 95:5 to 50:50) then yielded the desired Heck reaction products. Some regioisomers could not be separated and reported as a mixture. Full details for compounds **6a**–**9j** are attached in the Appendix A.

*(E)-N-(2-(3-(4-Methoxyphenyl)prop-1-en-1-yl)phenyl)acetamide* **6a**. IR (neat) ν: 3286 (m, NH), 3004 (w), 2833 (w), 1653 (s, C=O), 1510 (s), 1453 (m), 1371 (m), 1273 (m), 1242 (s), 1179 (m), 1035 (m, C−O), 821 (m), 753 (s) cm^−1^; ^1^H NMR (CDCl_3_, 500 MHz) δ*:* 7.76 (d, *J* = 8.0 Hz, H-2, 1H), 7.41 (br s, NH, 1H), 7.38 (d, *J* = 8.0 Hz, H-5, 1H), 7.22 (dd, *J =* 8.0, 7.5 Hz, H-3, 1H), 7.16 (br d, *J =* 8.5 Hz, H-11, 2H), 7.11 (dd, *J =* 8.0, 7.5 Hz, H-4, 1H), 6.87 (br d, *J =* 8.5 Hz, H-12, 2H), 6.45 (d, *J =* 15.5 Hz, H-7, 1H), 6.25 (dt, *J =* 15.5, 7.0 Hz, H-8, 1H), 3.80 (s, H-14, 3H), 3.51 (d, *J =* 7.0 Hz, H-9, 2H), 2.14 (s, H-16, 3H); ^13^C NMR (CDCl_3_, 125.8 MHz) δ*:* 168.4 (C-15), 158.1 (C-13), 134.1 (C-1), 133.9 (C-8), 131.7 (C-10), 130.2 (C-6), 129.5 (C-11), 127.8 (C-3), 126.9 (C-5), 125.6 (C-7), 125.3 (C-4), 123.8 (C-2), 114.00 (C-12), 55.2 (C-14), 38.6 (C-9), 24.2 (C-16). HRMS (+ESI) [M + H]^+^: 282.1475, C_18_H_20_NO_2_ requires 282.1494.

*(E)-N-(2-(3-(4-Methoxyphenyl)prop-1-en-1-yl)phenyl)isobutyramide* **6b**. IR (neat) ν: 3268 (w, NH), 3032 (w), 2969 (w), 1651 (m, C=O), 1511 (s), 1452 (m), 1382 (w), 1287 (w), 1178 (w), 1033 (m, C−O), 834 (w), 746 (s) cm^−1^; ^1^H NMR (CDCl_3_, 500 MHz) δ*:* 7.86 (d, *J* = 8.0 Hz, H-2, 1H), 7.38 (br s, NH, 1H), 7.35 (d, *J* = 8.0 Hz, H-5, 1H), 7.22 (td, *J* = 8.0, 1.5 Hz, H-3, 1H), 7.15 (d, *J* = 8.5 Hz, H-11, 2H), 7.08–7.13 (m, H-4, 1H), 6.87 (d, *J =* 8.5 Hz, H-12, 2H), 6.40 (d, *J =* 15.0 Hz, H-7, 1H), 6.26 (dt, *J =* 15.0, 6.6 Hz, H-8, 1H), 3.80 (s, H-14, 3H), 3.51 (d, *J =* 6.6 Hz, H-9, 2H), 2.47 (sept, *J =* 6.8 Hz, H-16, 1H), 1.21 (d, *J =* 6.8 Hz, H-17, 6H); ^13^C NMR (CDCl_3_, 125.8 MHz) δ*:* 175.0 (C-15), 158.2 (C-13), 134.5 (C-8), 134.2 (C-1), 131.5 (C-10), 129.8 (C-6), 129.6 (C-11), 127.8 (C-3), 127.1 (C-5), 125.5 (C-7), 124.9 (C-4), 123.2 (C-2), 114.0 (C-12), 55.3 (C-14), 38.6 (C-9), 36.6 (C-16), 19.6 (C-17). HRMS (+ESI) [M + H]^+^: 310.1823, C_20_H_24_NO_2_ requires 310.1807.

*(E)-N-(2-(3-(4-Methoxyphenyl)prop-1-en-1-yl)phenyl)butyramide* **6c**. IR (neat) ν: 3290 (w, NH), 3005 (w), 2959 (w), 1648 (s, C=O), 1511 (s), 1453 (m), 1381 (w), 1291 (m), 1175 (m), 1159, 1037 (m, C−O), 829 (w), 747 (s) cm^−1^; ^1^H NMR (CDCl_3_, 500 MHz) δ*:* 7.84 (d, *J* = 7.6 Hz, H-2, 1H), 7.36 (d, *J =* 7.6 Hz, H-5, 1H), 7.34 (br s, NH, 1H), 7.16 (d, *J =* 8.2 Hz, H-11, 2H), 7.10–7.12 (m, H-3, H-4, 2H), 6.87 (d, *J =* 8.2 Hz, H-12, 2H), 6.41 (d, *J =* 15.3 Hz, H-7, 1H), 6.25 (dt, *J =* 15.3, 6.5 Hz, H-8, 1H), 3.80 (s, H-14, 3H), 3.52 (d, *J =* 6.5 Hz, H-9, 2H), 2.30 (t, *J =* 7.4 Hz, H-16, 2H), 1.67–1.72 (m, H-17, 2H), 0.96 (t, *J =* 7.4 Hz, H-18, 3H); ^13^C NMR (CDCl_3_, 125.8 MHz) δ*:* 171.2 (C-15), 158.2 (C-13), 134.4 (C-8), 134.2 (C-1), 131.6 (C-10), 129.8 (C-6), 129.6 (C-11), 127.8 (C-3), 127.1 (C-5), 125.5 (C-7), 125.1 (C-4), 123.4 (C-2), 114.0 (C-12), 55.3 (C-14), 39.5 (C-16), 38.6 (C-9), 18.7 (C-17), 13.5 (C-18). HRMS (+ESI) [M + H]^+^: 310.1816, C_20_H_24_NO_2_ requires 310.1807.

*(E)-N-(2-(3-(4-Methoxyphenyl)prop-1-en-1-yl)phenyl)pentanamide* **6d**. IR (neat) ν: 3262 (w, NH), 3002 (w), 2959 (w), 1652 (s, C=O), 1509 (s), 1450 (m), 1380 (w), 1292 (w), 1174 (m), 1037 (m, C−O), 813 (w), 747 (s) cm−1; ^1^H NMR (CDCl_3_, 500 MHz) δ*:* 7.88 (d, *J* = 7.8 Hz, H-2, 1H), 7.38 (br s, NH, 1H), 7.37 (d, *J* = 7.8 Hz, H-5, 1H), 7.27−7.31 (m, H-3, 1H), 7.18 (d, *J* = 8.5 Hz, H-11, 2H), 7.10−7.16 (m, H-4, 1H), 6.89 (d, *J* = 8.5 Hz, H-12, 2H), 6.44 (d, *J* = 15.2 Hz, H-7, 1H), 6.27 (dt, *J* = 15.2, 6.3 Hz, H-8, 1H), 3.82 (s, H-14, 3H), 3.54 (d, *J* = 6.3 Hz, H-9, 2H), 2.34 (t, *J* = 7.5 Hz, H-16, 2H), 1.64−1.73 (m, H-17, 2H), 1.40−1.46 (m, H-18, 2H), 0.97 (t, *J* = 7.5 Hz, H-19, 3H); ^13^C NMR (CDCl_3_, 125.8 MHz) δ*:* 171.3 (C-15), 158.2 (C-13), 134.3 (C-8), 134.2 (C-1), 131.6 (C-10), 129.8 (C-6), 129.5 (C-11), 127.8 (C-3), 127.1 (C-5), 125.6 (C-7), 125.0 (C-4), 123.4 (C-2), 114.0 (C-12), 55.3 (C-14), 38.7 (C-9), 37.4 (C-16), 27.8 (C-17), 22.3 (C-18), 13.7 (C-19). HRMS (+ESI) [M + H]^+^: 324.1958, C_21_H_26_NO_2_ requires 324.1964.

*(E)-N-(2-(3-(4-Methoxyphenyl)prop-1-en-1-yl)phenyl)hexanamide* **6e**. IR (neat) ν: 3278 (w, NH), 2952 (w), 2929 (w), 1651 (s, C=O), 1510 (s), 1451 (m), 1378 (w), 1296 (m), 1176 (m), 1037 (m, C−O), 824 (w), 747 (s) cm^−1^; ^1^H NMR (CDCl_3_, 500 MHz) δ*:* 7.85 (d, *J* = 7.5 Hz, H-2, 1H), 7.36 (br s, NH, 1H), 7.35 (d, *J* = 7.5 Hz, H-5, 1H), 7.27−7.30 (m, H-3, 1H), 7.16 (d, *J* = 8.5 Hz, H-11, 2H), 7.09−7.13 (m, H-4, 1H), 6.87 (d, *J* = 8.5 Hz, H-12, 2H), 6.43 (d, *J* = 15.0 Hz, H-7, 1H), 6.24 (dt, *J* = 15.0, 6.8 Hz, H-8, 1H), 3.80 (s, H-14, 3H), 3.52 (d, *J* = 6.8 Hz, H-9, 2H), 2.29 (t, *J* = 7.2 Hz, H-16, 2H), 1.63−1.71 (m, H-17, 2H), 1.34−1.36 (m, H-18, 2H), 1.27−1.29 (m, H-19, 2H), 0.85 (t, *J* = 7.2 Hz, H-20, 3H); ^13^C NMR (CDCl_3_, 125.8 MHz) δ*:* 171.3 (C-15), 158.2 (C-13), 134.4 (C-8), 134.3 (C-1), 131.6 (C-10), 129.8 (C-6), 129.5 (C-11), 127.8 (C-3), 127.1 (C-5), 125.6 (C-7), 124.99 (C-4), 123.4 (C-2), 113.99 (C-12), 55.2 (C-14), 38.7 (C-9), 37.8 (C-16), 31.4 (C-18), 25.39 (C-17), 22.4 (C-19), 13.93 (C-20). HRMS (+ESI) [M + H]^+^: 338.2103, C_22_H_28_NO_2_ requires 338.2120.

*(E)-N-(2-(3-(4-Methoxyphenyl)prop-1-en-1-yl)phenyl)decanamide* **6f**. IR (neat) ν: 3297 (w, NH), 2954 (w), 2920 (m), 1648 (m, C=O), 1509 (s), 1453 (m), 1378 (w), 1293 (m), 1174 (m), 1034 (m, C−O), 827 (w), 748 (s) cm^−1^; ^1^H NMR (CDCl_3_, 500 MHz) δ*:* 7.82 (d, *J* = 7.7 Hz, H-2, 1H), 7.42 (br s, NH, 1H), 7.36 (d, *J* = 7.7 Hz, H-5, 1H), 7.21−7.25 (m, H-3, 1H), 7.16 (d, *J* = 8.5 Hz, H-11, 2H), 7.08−7.13 (m, H-4, 1H), 6.87 (d, *J* = 8.5 Hz, H-12, 2H), 6.44 (d, *J* = 15.5 Hz, H-7, 1H), 6.24 (dt, *J* = 15.5, 6.5 Hz, H-8, 1H), 3.80 (s, H-14, 3H), 3.51 (d, *J* = 6.5 Hz, H-9, 2H), 2.27−2.33 (m, H-16, 2H), 1.62−1.70 (m, H-17, 2H), 1.15−1.28 (m, H-18, H-19, H-20, H-21, H-22, H-23, 12H), 0.89 (t, *J* = 6.9 Hz, H-24, 3H); ^13^C NMR (CDCl_3_, 125.8 MHz) δ*:* 171.4 (C-15), 158.2 (C-13), 134.3 (C-1), 134.1 (C-8), 131.6 (C-10), 130.1 (C-6), 129.5 (C-11), 127.8 (C-3), 127.0 (C-5), 125.6 (C-7), 125.0 (C-4), 123.5 (C-2), 113.97 (C-12), 55.2 (C-14), 38.6 (C-9), 37.8 (C-16), 31.81 (C-22), 29.4 (C-21), 29.3 (C-20), 29.2 (C-18, C-19), 25.75 (C-17), 22.6 (C-23), 14.0 (C-24). HRMS (+ESI) [M + H]^+^: 394.2741, C_26_H_36_NO_2_ requires 394.2746.

*(E)-N-(2-(3-(4-Methoxyphenyl)prop-1-en-1-yl)phenyl)cyclohexanecarboxamide* **6g**. IR (neat) ν: 3260 (w, NH), 3034 (w), 2931 (w), 1652 (s, C=O), 1511 (s), 1450 (m), 1384 (w), 1300 (w), 1175 (w), 1039 (m, C−O), 824 (w), 747 (s) cm^−1^; ^1^H NMR (CDCl_3_, 500 MHz) δ*:* 7.88 (d, *J* = 7.8 Hz, H-2, 1H), 7.38 (br s, NH, 1H), 7.34 (d, *J* = 7.8 Hz, H-5, 1H), 7.23 (t, *J* = 7.8 Hz, H-3, 1H), 7.16 (d, *J* = 8.6 Hz, H-11, 2H), 7.11 (t, *J* = 7.8 Hz, H-4, 1H), 6.88 (d, *J* = 8.6 Hz, H-12, 2H), 6.43 (d, *J* = 15.7 Hz, H-7, 1H), 6.27 (dt, *J* = 15.7, 5.8 Hz, H-8, 1H), 3.80 (s, H-14, 3H), 3.53 (d, *J* = 5.8 Hz, H-9, 2H), 2.18 (td, *J* = 11.7, 2.7 Hz, H-16, 1H), 1.92 (d, *J* = 12.6 Hz, H-17α, 2H), 1.75−1.83 (m, H-17β, 2H), 1.65–1.72 (m, H-19β, 1H), 1.40−1.49 (m, H-18α, 2H), 1.15−1.34 (m, H-18β, H-19α, 3H); ^13^C NMR (CDCl_3_, 125.8 MHz) δ*:* 174.1 (C-15), 158.2 (C-13), 134.5 (C-8), 134.3 (C-1), 131.5 (C-10), 130.1 (C-6), 129.7 (C-11), 127.8 (C-3), 127.1 (C-5), 125.6 (C-7), 124.8 (C-4), 123.1 (C-2), 114.0 (C-12), 55.2 (C-14), 46.4 (C-16), 38.6 (C-9), 29.7 (C-17), 25.7 (C-18, C-19). HRMS (+ESI) [M + H]^+^: 350.2123, C_23_H_28_NO_2_ requires 350.2120.

*(E)-N-(2-(3-(4-Methoxyphenyl)prop-1-en-1-yl)phenyl)benzamide* **6h**. IR (neat) ν: 3259 (w, NH), 3030 (w), 2834 (w), 1647 (s, C=O), 1512 (s), 1483 (s), 1440 (w), 1300 (m), 1176 (m), 1037 (m, C−O), 815 (w), 753 (s) cm^−1^; ^1^H NMR (CDCl_3_, 500 MHz) δ*:* 7.98 (d, *J* = 7.9 Hz, H-2, 1H), 7.75 (br s, NH, 1H), 7.70 (d, *J* = 7.4 Hz, H-17, 2H), 7.49 (t, *J* = 7.4 Hz, H-19, 1H), 7.41 (d, *J* = 7.9 Hz, H-5, 1H), 7.29 (t, *J* = 7.4 Hz, H-18, 2H), 7.26–7.33 (m, H-3, 1H), 7.08 (d, *J* = 8.7 Hz, H-11, 2H), 7.06–7.09 (m, H-4, 1H), 6.78 (d, *J* = 8.7 Hz, H-12, 2H), 6.39 (d, *J* = 15.7 Hz, H-7, 1H), 6.26 (dt, *J* = 15.7, 6.0 Hz, H-8, 1H), 3.72 (s, H-14, 3H), 3.45 (d, *J* = 6.0 Hz, H-9, 2H); ^13^C NMR (CDCl_3_, 125.8 MHz) δ: 165.4 (C-15), 158.1 (C-13), 134.9 (C-8, C-16), 134.3 (C-1), 131.8 (C-19), 131.4 (C-10), 130.02 (C-6), 129.7 (C-11), 128.8 (C-18), 127.9 (C-3), 127.3 (C-5), 127.1 (C-17), 125.6 (C-7), 125.1 (C-4), 123.1 (C-2), 114.0 (C-12), 55.3 (C-14), 38.6 (C-9). HRMS (+ESI) [M + H]^+^: 344.1674, C_23_H_22_NO_2_ requires 344.1651.

*(E)-N-(2-(3-(4-Methoxyphenyl)prop-1-en-1-yl)phenyl)-2-methylbenzamide* **6i**. IR (neat) ν: 3275 (w, NH), 3027 (w), 2997 (w), 1646 (s, C=O), 1514 (s), 1444 (m), 1306 (m), 1271 (m), 1249, 1177 (m), 1037 (m, C−O), 827 (w), 748 (s) cm^−1^; ^1^H NMR (CDCl_3_, 500 MHz) δ*:* 8.02 (d, *J* = 7.5 Hz, H-2, 1H), 7.76 (br s, NH, 1H), 7.43−7.35 (m, H-5, H-19, H-21, 3H), 7.31−7.26 (m, H-3, H-18, 2H), 7.18−7.13 (m, H-4, H-20, 2H), 7.11 (d, *J* = 8.5 Hz, H-11, 2H), 6.82 (d, *J* = 8.5 Hz, H-12, 2H), 6.44 (br d, *J* = 15.5 Hz, H-7, 1H), 6.29 (dt, *J* = 15.5, 6.5 Hz, H-8, 1H), 3.79 (s, H-14, 3H), 3.50 (br d, *J* = 6.5 Hz, H-9, 2H), 2.51 (s, H-22, 3H); ^13^C NMR (CDCl_3_, 125.8 MHz) δ: 168.0 (C-15), 158.1 (C-13), 136.54 (C-16), 136.3 (C-17), 134.6 (C-8), 134.2 (C-1), 131.36 (C-10), 131.28 (C-19), 130.3 (C-6), 130.20 (C-18), 129.6 (C-11), 127.9 (C-3), 127.2 (C-5), 126.57 (C-21), 125.9 (C-20), 125.5, 125.4 (C-4, C-7), 123.3 (C-2), 114.0 (C12), 55.2 (C-14), 38.5 (C-9), 19.9 (C-22). HRMS (+ESI) [M + H]^+^: 358.1801, C_24_H_24_NO_2_ requires 358.1807.

*(E)-N-(2-(3-(4-Methoxyphenyl)prop-1-en-1-yl)phenyl)furan-2-carboxamide* **6j**. IR (neat) ν: 3355 (w, NH), 3126 (w), 2934 (w), 1669 (m, C=O), 1510 (s), 1450 (m), 1301 (m), 1164 (m), 1010 (m, C−O), 835 (w), 753 (s) cm^−1^; ^1^H NMR (CDCl_3_, 500 MHz) δ*:* 8.44 (br s, NH, 1H), 8.10 (d, *J* = 8.2 Hz, H-19, 1H), 7.26–7.31 (m, H-2, H-3, H-5, 3H), 7.18 (d, *J* = 7.5 Hz, H-11, 2H), 7.12–7.15 (m, H-4, H-17, 2H), 6.86 (d, *J* = 7.5 Hz, H-12, 2H), 6.58 (d, *J* = 15.8 Hz, H-7, 1H), 6.54–6.57 (m, H-18, 1H), 6.31 (dt, *J* = 15.8, 6.2 Hz, H-8, 1H), 3.80 (s, H-14, 3H), 3.55 (d, *J* = 6.2 Hz, H-7, 2H); ^13^C NMR (CDCl_3_, 125.8 MHz) δ*:* 158.1 (C-13), 156.1 (C-15), 148.0 (C-16), 144.2 (C-19), 134.8 (C-8), 133.7 (C-1), 131.5 (C-10), 129.7 (C-11), 129.48 (C-6), 127.9 (C-3), 125.5 (C-5), 125.1 (C-7), 125.0 (C-4), 122.6 (C-2), 115.12 (C-17), 114.0 (C-12), 112.5 (C-18), 55.2 (C-14), 38.7 (C-9). HRMS (+ESI) [M + H]^+^: 334.1445, C_21_H_20_NO_3_ requires 334.1443.

*(E)-N-(2-(3-(4-Methoxyphenyl)allyl)phenyl)acetamide* **7a**. IR (neat) ν: 3286 (m, NH), 3004 (w), 2833 (w), 1653 (s, C=O), 1510 (s), 1453 (m), 1371 (m), 1273 (m), 1242 (s), 1179 (m), 1035 (m, C−O), 821 (m), 753 (s) cm^−1^; ^1^H NMR (CDCl_3_, 500 MHz) δ*:* 7.82 (d, *J* = 8.0 Hz, H-2′, 1H), 7.28 (br d, *J* = 8.5 Hz, H-11′, 2H), 7.27 (dd, *J =* 8.0, 7.5 Hz, H-3′, 1H), 7.26–7.29 (m, NH, 1H), 7.23 (d, *J* = 8.0 Hz, H-5′, 1H), 7.13 (dd, *J =* 8.0, 7.5 Hz, H-4′, 1H), 6.84 (br d, *J =* 8.5 Hz, H-12′, 2H), 6.40 (d, *J =* 16.0 Hz, H-9′, 1H), 6.17 (dt, *J* = 16.0, 6.5 Hz, H-8′, 1H), 3.80 (s, H-14′, 3H), 3.51 (d, *J =* 6.5 Hz, H-7′, 2H), 2.11 (s, H-16′, 3H); ^13^C NMR (CDCl_3_, 125.8 MHz) δ*:* 168.3 (C-15′), 159.2 (C-13′), 136.0 (C-1′), 131.1 (C-9′), 130.6 (C-6′), 130.1 (C-5′), 129.4 (C-10′), 127.4 (C-3′), 127.3 (C-11′), 125.4 (C-8′), 125.3 (C-4′), 123.9 (C-2′), 113.96 (C-12′), 55.2 (C-14′), 36.0 (C-7′), 24.3 (C-16′). HRMS (+ESI) [M + H]^+^: 282.1475, C_18_H_20_NO_2_ requires 282.1494.

*(E)-N-(2-(3-(4-Methoxyphenyl)allyl)phenyl)isobutyramide* **7b**. IR (neat) ν: 3268 (w, NH), 3032 (w), 2969 (w), 1651 (m, C=O), 1511 (s), 1452 (m), 1382 (w), 1287 (w), 1178 (w), 1033 (m, C−O), 834 (w), 746 (s) cm−1; ^1^H NMR (CDCl_3_, 500 MHz) δ*:* 7.94 (d, *J* = 7.5 Hz, H-2′, 1H), 7.38 (br s, NH, 1H), 7.24–7.27 (m, H-3′, H-11′, 3H), 7.23 (d, *J* = 7.5 Hz, H-5′, 1H), 7.08–7.13 (m, H-4′, 1H), 6.84 (d, *J =* 8.7 Hz, H-12′, 2H), 6.38 (d, *J =* 15.8 Hz, H-9′, 1H), 6.18 (dt, *J =* 15.8, 6.5 Hz, H-8′, 1H), 3.80 (s, H-14′, 3H), 3.52 (d, *J =* 6.5 Hz, H-7′, 2H), 2.47 (sept, *J =* 6.8 Hz, H-16′, 1H), 1.20 (d, *J =* 6.8 Hz, H-17′, 6H); ^13^C NMR (CDCl_3_, 125.8 MHz) δ*:* 175.1 (C-15′), 159.2 (C-13′), 135.6 (C-1′), 131.2 (C-9′), 130.2 (C-5′), 130.1 (C-6′), 129.4 (C-10′), 127.5 (C-3′), 127.3 (C-11′), 125.3 (C-8′), 125.0 (C-4′), 123.3 (C-2′), 114.0 (C-12′), 55.3 (C-14′), 36.5 (C-16′), 36.1 (C-7′), 19.6 (C-17′). HRMS (+ESI) [M + H]^+^: 310.1823, C_20_H_24_NO_2_ requires 310.1807.

*(E)-N-(2-(3-(4-Methoxyphenyl)allyl)phenyl)butyramide* **7c**. IR (neat) ν: 3290 (w, NH), 3005 (w), 2959 (w), 1648 (s, C=O), 1511 (s), 1453 (m), 1381 (w), 1291 (m), 1175 (m), 1159, 1037 (m, C−O), 829 (w), 747 (s) cm^−1^; ^1^H NMR (CDCl_3_, 500 MHz) δ*:* 7.91 (d, *J* = 8.0 Hz, H-2′, 1H), 7.34 (br s, NH, 1H), 7.28 (br dd, *J =* 8.0, 7.5 Hz, H-3′, 1H), 7.27 (br d, *J =* 8.5 Hz H-11′, 2H), 7.23 (d, *J =* 7.5 Hz, H-5′, 1H), 7.12 (t, *J =* 7.5 Hz, H-4′, 1H), 6.85 (br d, *J =* 8.5 Hz H-12′, 2H), 6.41 (br d, *J =* 16.0 Hz, H-9′, 1H), 6.18 (dt, *J =* 16.0, 6.0 Hz, H-8′, 1H), 3.80 (s, H-14′, 3H), 3.52 (br d, *J =* 6.0 Hz, H-7′, 2H), 2.28 (t, *J =* 7.5 Hz, H-16′, 2H), 1.70 (sext, *J* = 7.5 Hz, H-17′, 2H), 0.95 (t, *J =* 7.5 Hz, H-18′, 3H); ^13^C NMR (CDCl_3_, 125.8 MHz) δ*:* 171.2 (C-15′), 159.2 (C-13′), 136.2 (C-1′), 131.2 (C-9′), 130.2 (C-5′), 130.0 (C-6′), 129.4 (C-10′), 127.5 (C-3′), 127.3 (C-11′), 125.3 (C-8′), 125.0 (C-4′), 123.5 (C-2′), 114.0 (C-12′), 55.3 (C-14′), 39.7 (C-16′), 36.2 (C-7′), 19.2 (C-17′), 13.7 (C-18′). HRMS (+ESI) [M + H]^+^: 310.1816, C_20_H_24_NO_2_ requires 310.1807.

*(E)-N-(2-(3-(4-Methoxyphenyl)allyl)phenyl)pentanamide* **7d**. IR (neat) ν: 3262 (w, NH), 3002 (w), 2959 (w), 1652 (s, C=O), 1509 (s), 1450 (m), 1380 (w), 1292 (w), 1174 (m), 1037 (m, C−O), 813 (w), 747 (s) cm^−1^; ^1^H NMR (CDCl_3_, 500 MHz) δ: 7.94 (d, *J* = 7.7 Hz, H-2′, 1H), 7.38 (br s, NH, 1H), 7.30 (d, *J* = 8.8 Hz, H-11′, 2H), 7.27–7.31 (m, H-3′, 1H), 7.25 (d, *J* = 7.7 Hz, H-5′, 1H), 7.10−7.16 (m, H-4′, 1H), 6.87 (d, *J* = 8.8 Hz, H-12′, 2H), 6.42 (d, *J* = 15.8 Hz, H-9′, 1H), 6.20 (dt, *J* = 15.8, 6.0 Hz, H-8′, 1H), 3.82 (s, H-14′, 3H), 3.54 (d, *J* = 6.0 Hz, H-7′, 2H), 2.32 (t, *J* = 7.5 Hz, H-16′, 2H), 1.64−1.73 (m, H-17′, 2H), 1.34−1.39 (m, H-18′, 2H), 0.90 (t, *J* = 7.5 Hz, H-19′, 3H); ^13^C NMR (CDCl_3_, 125.8 MHz) δ: 171.3 (C-15′), 159.2 (C-13′), 136.3 (C-1′), 131.2 (C-9′), 130.2 (C-5′), 129.9 (C-6′), 129.4 (C-10′), 127.5 (C-3′), 127.3 (C-11′), 125.3 (C-8′), 125.0 (C-4′), 123.3 (C-2′), 114.0 (C-12′), 55.3 (C-14′), 37.6 (C-16′), 36.2 (C-7′), 27.8 (C-17′), 22.4 (C-18′), 13.8 (C-19′). HRMS (+ESI) [M + H]^+^: 324.1958, C_21_H_26_NO_2_ requires 324.1964.

*(E)-N-(2-(3-(4-Methoxyphenyl)allyl)phenyl)hexanamide* **7e**. IR (neat) ν: 3278 (w, NH), 2952 (w), 2929 (w), 1651 (s, C=O), 1510 (s), 1451 (m), 1378 (w), 1296 (m), 1176 (m), 1037 (m, C−O), 824 (w), 747 (s) cm^−1^; ^1^H NMR (CDCl_3_, 500 MHz) δ: 7.92 (d, *J* = 7.8 Hz, H-2′, 1H), 7.36 (br s, NH, 1H), 7.27−7.30 (m, H-3′, 1H), 7.27 (d, *J* = 8.8 Hz, H-11′, 2H), 7.22 (d, *J* = 7.8 Hz, H-5′, 1H), 7.09−7.13 (m, H-4′, 1H), 6.84 (d, *J* = 8.5 Hz, H-12′, 2H), 6.40 (d, *J* = 15.5 Hz, H-9′, 1H), 6.18 (dt, *J* = 15.5, 6.5 Hz, H-8′, 1H), 3.80 (s, H-14′, 3H), 3.52 (d, *J* = 6.5 Hz, H-7′, 2H), 2.29 (t, *J* = 7.2 Hz, H-16′, 2H), 1.63−1.71 (m, H-17′, 2H), 1.34−1.36 (m, H-18′, 2H), 1.27−1.29 (m, H-19′, 2H), 0.85 (t, *J* = 7.2 Hz, H-20′, 3H); ^13^C NMR (CDCl_3_, 125.8 MHz) δ*:* 171.3 (C-15′), 159.2 (C-13′), 136.3 (C-1′), 131.2 (C-9′), 130.3 (C-5′), 130.2 (C-6′), 129.4 (C-10′), 127.5 (C-3′), 127.3 (C-11′), 125.3 (C-8′), 125.01 (C-4′), 123.3 (C-2′), 114.02 (C-12′), 55.3 (C-14′), 37.6 (C-16′), 36.2 (C-7′), 31.4 (C-18′), 25.43 (C-17′), 22.3 (C-19′), 13.87 (C-20′). HRMS (+ESI) [M + H]^+^: 338.2103, C_22_H_28_NO_2_ requires 338.2120.

*(E)-N-(2-(3-(4-Methoxyphenyl)allyl)phenyl)decanamide* **7f**. IR (neat) ν: 3297 (w, NH), 2954 (w), 2920 (m), 1648 (m, C=O), 1509 (s), 1453 (m), 1378 (w), 1293 (m), 1174 (m), 1034 (m, C−O), 827 (w), 748 (s) cm^−1^; ^1^H NMR (CDCl_3_, 500 MHz) δ*:* 7.90 (d, *J* = 7.5 Hz, H-2′, 1H), 7.42 (br s, NH, 1H), 7.27 (d, *J* = 8.6 Hz, H-11′, 2H), 7.23 (d, *J* = 7.5 Hz, H-5′, 1H), 7.21−7.25 (m, H-3′, 1H), 7.08−7.13 (m, H-4′, 1H), 6.84 (d, *J* = 8.5 Hz, H-12′, 2H), 6.41 (d, *J* = 16.0 Hz, H-9′, 1H), 6.18 (dt, *J* = 16.0, 6.6 Hz, H-8′, 1H), 3.80 (s, H-14′, 3H), 3.51 (d, *J* = 6.6 Hz, H-7′, 2H), 2.27−2.33 (m, H-16′, 2H), 1.62−1.70 (m, H-17′, 2H), 1.15−1.28 (m, H-18′, H-19′, H-20′, H-21′, H-22′, H-23′, 12H), 0.89 (t, *J* = 6.9 Hz, H-24′, 3H); ^13^C NMR (CDCl_3_, 125.8 MHz) δ*:* 171.4 (C-15′), 159.2 (C-13′), 136.2 (C-1′), 131.2 (C-9′), 130.1 (C-5′), 129.9 (C-6′), 129.4 (C-10′), 127.4 (C-3′), 127.2 (C-11′), 125.3 (C-8′), 125.0 (C-4′), 123.5 (C-2′), 113.99 (C-12′), 55.2 (C-14′), 37.5 (C-16′), 36.1 (C-7′), 31.79 (C-22′), 29.4 (C-21′), 29.3 (C-20′), 29.2 (C-18′, C-19′), 25.70 (C-17′), 22.6 (C-23′), 14.0 (C-24′). HRMS (+ESI) [M + H]^+^: 394.2741, C_26_H_36_NO_2_ requires 394.2746.

*(E)-N-(2-(3-(4-Methoxyphenyl)allyl)phenyl)cyclohexanecarboxamide* **7g**. IR (neat) ν: 3260 (w, NH), 3034 (w), 2931 (w), 1652 (s, C=O), 1511 (s), 1450 (m), 1384 (w), 1300 (w), 1175 (w), 1039 (m, C−O), 824 (w), 747 (s) cm^−1^; ^1^H NMR (CDCl_3_, 500 MHz) δ*:* 7.97 (d, *J* = 7.6 Hz, H-2′, 1H), 7.38 (br s, NH, 1H), 7.28 (d, *J* = 8.8 Hz, H-11′, 2H), 7.24−7.28 (m, H-3′, 1H), 7.21−7.24 (m, H-5′, 1H), 7.09 (t, *J* = 7.6 Hz, H-4′, 1H), 6.85 (d, *J* = 8.8 Hz, H-12′, 2H), 6.36 (d, *J* = 15.8 Hz, H-9′, 1H), 6.18 (dt, *J* = 15.8, 6.0 Hz, H-8′, 1H), 3.81 (s, H-14′, 3H), 3.52 (d, *J* = 6.0 Hz, H-7′, 2H), 2.18 (td, *J* = 11.7, 2.7 Hz, H-16′, 1H), 1.92 (d, *J* = 12.6 Hz, H-17α′, 2H), 1.75−1.83 (m, H-17β′, 2H), 1.65–1.72 (m, H-19β′, 1H), 1.40−1.49 (m, H-18α′, 2H), 1.15−1.34 (m, H-18β′, H-19α′, 3H); ^13^C NMR (CDCl_3_, 125.8 MHz) δ*:* 174.1 (C-15′), 159.2 (C-13′), 136.4 (C-1′), 131.4 (C-9′), 130.2 (C-5′), 130.1 (C-6′), 129.3 (C-10′), 127.5 (C-3′), 127.3 (C-11′), 125.3 (C-8′), 124.8 (C-4′), 123.2 (C-2′), 114.1 (C-12′), 55.3 (C-14′), 46.5 (C-16′), 36.3 (C-7′), 29.8 (C-17′), 25.7 (C-18′, C-19′). HRMS (+ESI) [M + H]^+^: 350.2123, C_23_H_28_NO_2_ requires 350.2120.

*(E)-N-(2-(3-(4-Methoxyphenyl)allyl)phenyl)benzamide* **7h**. IR (neat) ν: 3259 (w, NH), 3030 (w), 2834 (w), 1647 (s, C=O), 1512 (s), 1483 (s), 1440 (w), 1300 (m), 1176 (m), 1037 (m, C−O), 815 (w), 753 (s) cm^−1^; ^1^H NMR (CDCl_3_, 500 MHz) δ*:* 8.16 (br s, NH, 1H), 8.04 (d, *J* = 7.7 Hz, H-2′, 1H), 7.70 (d, *J* = 7.4 Hz, H-17′, 2H), 7.43 (t, *J* = 7.4 Hz, H-19′, 1H), 7.29 (t, *J* = 7.4 Hz, H-18′, 2H), 7.26–7.33 (m, H-3′, 1H), 7.22 (d, *J* = 8.7 Hz, H-11′, 2H), 7.20–7.23 (m, H-5′, 1H), 7.10 (td, *J* = 7.7, 1.2 Hz, H-4′, 1H), 6.80 (d, *J* = 8.7 Hz, H-12′, 2H), 6.42 (d, *J* = 16.0 Hz, H-9′, 1H), 6.17 (dt, *J* = 16.0, 6.0 Hz, H-8′, 1H), 3.76 (s, H-14′, 3H), 3.53 (d, *J* = 6.0 Hz, H-7′, 2H); ^13^C NMR (CDCl_3_, 125.8 MHz) δ: 165.4 (C-15′), 159.3 (C-13′), 136.6 (C-1′), 134.8 (C-16′), 134.8 (C-8′), 131.7 (C-9′, C-19′), 130.3 (C-5′), 129.99 (C-6′), 129.1 (C-10′), 128.7 (C-18′), 127.7 (C-3′), 127.4 (C-11′), 127.0 (C-17′), 125.2 (C-8′), 125.1 (C-4′), 123.1 (C-2′), 114.1 (C12′), 55.2 (C-14′), 36.8 (C-7′). HRMS (+ESI) [M + H]^+^: 344.1674, C_23_H_22_NO_2_ requires 344.1651.

*(E)-N-(2-(3-(4-Methoxyphenyl)allyl)phenyl)-2-methylbenzamide* **7i**. IR (neat) ν: 3275 (w, NH), 3027 (w), 2997 (w), 1646 (s, C=O), 1514 (s), 1444 (m), 1306 (m), 1271 (m), 1249, 1177 (m), 1037 (m, C−O), 827 (w), 748 (s) cm^−1^; ^1^H NMR (CDCl_3_, 500 MHz) δ*:* 8.16 (d, *J* = 7.5 Hz, H-2′, 1H), 7.75 (br s, NH, 1H), 7.37 (br d, *J* = 8.0 Hz, H-21′, 1H), 7.34 (dd, *J* = 7.5, 7.0 Hz, H-19′, 1H), 7.33 (td, *J* = 7.5, 1.0 Hz, H-3′, 1H), 7.27 (dd, *J* = 7.5, 1.0 Hz, H-5′, 1H), 7.23 (d, *J* = 7.5 Hz, H-18′, 1H), 7.17 (br d, *J* = 8.5 Hz, H-11′, 2H), 7.16 (ddd, *J* = 8.0, 7.0, 1.0, H-20′, 1H), 7.08 (t, *J* = 7.5 Hz, H-4′, 1H), 6.82 (br d, *J* = 8.5 Hz, H-12′, 2H), 6.30 (dt, *J* = 16.0, 1.5 Hz, H-9′, 1H), 6.16 (dt, *J* = 16.0, 6.5 Hz, H-8′, 1H), 3.81 (s, H-14′, 3H), 3.55 (dd, *J* = 6.5, 1.5 Hz, H-7′, 2H), 2.48 (s, H-22′, 3H); ^13^C NMR (CDCl_3_, 125.8 MHz) δ: 167.9 (C-15′), 159.2 (C-13′), 136.7 (C-1′), 136.48 (C-16′), 136.2 (C-17′), 131.5 (C-19′), 131.33 (C-9′), 130.3, 130.17 (C-5′, C18′), 129.9 (C-6′), 129.3 (C-10′), 127.6 (C-3′), 127.3 (C-11′), 126.65 (C-21′), 125.8 (C-20′), 125.2 (C-8′), 125.0 (C-4′), 122.9 (C-2′), 113.9 (C12′), 55.3 (C-14′), 36.3 (C-7′), 19.9 (C-22′). HRMS (+ESI) [M + H]^+^: 358.1801, C_24_H_24_NO_2_ requires 358.1807.

*(E)-N-(2-(3-(4-Methoxyphenyl)allyl)phenyl)furan-2-carboxamide* **7j**. IR (neat) ν: 3355 (w, NH), 3126 (w), 2934 (w), 1669 (m, C=O), 1510 (s), 1450 (m), 1301 (m), 1164 (m), 1010 (m, C−O), 835 (w), 753 (s) cm^−1^; ^1^H NMR (CDCl_3_, 500 MHz) δ*:* 8.15 (br s, NH, 1H), 8.10 (d, *J* = 8.2 Hz, H-19′, 1H), 7.39 (d, *J* = 7.5 Hz, H-2′, 1H), 7.29 (d, *J* = 7.6 Hz, H-11′, 2H), 7.26–7.31 (m, H-3′, H-5′, 2H), 7.12–7.15 (m, H-4′, H-17′, 2H), 6.85 (d, *J* = 7.6 Hz, H-12′, 2H), 6.54–6.57 (m, H-18′, 1H), 6.52 (d, *J* = 15.5 Hz, H-9′, 1H), 6.20 (dt, *J* = 15.5, 6.1 Hz, H-8′, 1H), 3.80 (s, H-14, 3H), 3.61 (d, *J* = 6.1 Hz, H-7′, 2H); ^13^C NMR (CDCl_3_, 125.8 MHz) δ*:* 159.2 (C-13′), 156.0 (C-15′), 148.0 (C-16′), 144.2 (C-19′), 135.9 (C-1′), 131.9 (C-9′), 130.2 (C-5′), 130.0 (C-6′), 129.54 (C-10′), 127.6 (C-3′), 127.3 (C-11′), 125.0 (C-8′), 124.8 (C-4′), 122.9 (C-2′), 115.08 (C-17′), 114.0 (C-12′), 112.4 (C-18′), 55.2 (C-14′), 36.6 (C-7′). HRMS (+ESI) [M + H]^+^: 334.1445, C_21_H_20_NO_3_ requires 334.1443.

*(E)-N-(2-(3-(4-Hydroxy-3-methoxyphenyl)prop-1-en-1-yl)phenyl)acetamide* **8a**. IR (neat) ν: 3348 (w, OH), 3229 (w, NH), 3125 (w), 2965 (w), 1636 (m, C=O), 1517 (s), 1460 (w), 1374 (w), 1273 (s), 1236 (m), 1175 (m), 1033 (m, C−O), 868 (w), 822 (w), 797 (w), 756 (s) cm^−1^; ^1^H NMR (CDCl_3_, 500 MHz) δ: 7.77 (d, *J* = 8.0 Hz, H-2, 1H), 7.38 (d, *J* = 7.5 Hz, H-5, 1H), 7.29 (br s, NH, 1H), 6.64–7.17 (m, H-3, H-4, H-11, H-14, H-15, 5H), 6.45 (br d, *J* = 15.5 Hz, H-7, 1H), 6.26 (dt, *J* = 15.5, 7.0 Hz, H-8, 1H), 5.56 (s, OH, 1H), 3.89 (s, H-16, 3H), 3.51 (d, *J* = 7.0 Hz, H-9, 2H), 2.15 (s, H-18, 3H); ^13^C NMR (CDCl_3_, 125.8 MHz) δ: 168.3 (C-17), 146.6 (C-12), 144.2 (C-13), 134.2 (C-1), 134.0 (C-8), 131.5 (C-10), 130.5 (C-6), 127.9 (C-3), 127.0 (C-5), 125.7 (C-7), 125.4 (C-4), 123.8 (C-2), 121.2 (C-15), 114.43 (C-14), 111.2 (C-11), 55.95 (C-16), 39.2 (C-9), 24.2 (C-18). HRMS (+ESI) [M + H]^+^: 298.1435, C_18_H_20_NO_3_ requires 298.1443.

*(E)-N-(2-(3-(4-Hydroxy-3-methoxyphenyl)prop-1-en-1-yl)phenyl)isobutyramide* **8b**. IR (neat) ν: 3398 (w, OH), 3291 (m, NH), 3016 (w), 2970 (w), 1654 (s, C=O), 1509 (s), 1446 (m), 1371 (w), 1273 (s), 1236 (s), 1155 (m), 1038 (m, C−O), 848 (m), 812 (w), 798 (w), 753 (s) cm^−1^; ^1^H NMR (CDCl_3_, 500 MHz) δ: 7.84 (d, *J* = 7.6 Hz, H-2, 1H), 7.43 (d, *J* = 7.6 Hz, H-5, 1H), 7.36 (br s, NH, 1H), 7.21 (td, *J* = 7.6, 1.7 Hz, H-3, 1H), 7.06 (t, *J* = 7.6 Hz, H-4, 1H), 6.72–6.74 (m, H-11, H-14, H-15, 3H), 6.42 (d, *J* = 15.8 Hz, H-7, 1H), 6.27 (dt, *J* = 15.8, 6.5 Hz, H-8, 1H), 5.61 (br s, OH, 1H), 3.88 (s, H-16, 3H), 3.50 (d, *J* = 6.5 Hz, H-9, 2H), 2.47 (sept, *J* = 7.0 Hz, H-18, 1H), 1.20 (d, *J* = 7.0 Hz, H-19, 6H); ^13^C NMR (CDCl_3_, 125.8 MHz) δ: 175.1 (C-17), 146.7 (C-13), 144.2 (C-12), 135.7 (C-1), 134.2 (C-8), 131.4 (C-10), 130.0 (C-6), 127.9 (C-3), 127.0 (C-5), 125.6 (C-7), 124.9 (C-4), 122.8 (C-2), 121.3 (C-15), 114.7 (C-14), 111.2 (C-11), 55.9 (C-16), 39.2 (C-9), 36.7 (C-18), 19.6 (C-19). HRMS (+ESI) [M + H]^+^: 326.1755, C_20_H_24_NO_3_ requires 326.1756.

*(E)-N-(2-(3-(4-Hydroxy-3-methoxyphenyl)prop-1-en-1-yl)phenyl)butyramide* **8c**. IR (neat) ν: 3399 (w, OH), 3269 (w, NH), 3053 (w), 2936 (w), 1646 (s, C=O), 1510 (s), 1453 (m), 1376 (w), 1271 (s), 1229 (s), 1155 (m), 1034 (m, C−O), 849 (w), 806 (w), 794 (w), 756 (s) cm^−1^; ^1^H NMR (CDCl_3_, 500 MHz) δ: 7.82 (d, *J* = 7.7 Hz, H-2, 1H), 7.32 (br s, NH, 1H), 7.37 (d, *J* = 7.7 Hz, H-5, 1H), 7.08–7.09 (m, H-3, 1H), 7.01 (t, *J* = 7.7 Hz, H-4, 1H), 6.73–6.75 (m, H-11, H-14, H-15, 3H), 6.42 (d, *J* = 15.3 Hz, H-7, 1H), 6.26 (dt, *J* = 15.3, 7.1 Hz, H-8, 1H), 5.57 (s, OH, 1H), 3.88 (s, H-16, 3H), 3.50 (d, *J* = 7.1 Hz, H-9, 2H), 2.29 (t, *J* = 7.3 Hz, H-18, 2H), 1.71 (sext, *J* = 7.3 Hz, H-19, 2H), 0.96 (t, *J* = 7.3 Hz, H-20, 3H); ^13^C NMR (CDCl_3_, 125.8 MHz) δ: 171.2 (C-17), 146.6 (C-13), 144.2 (C-12), 134.2 (C-1), 134.1 (C-8), 131.4 (C-10), 130.0 (C-6), 127.9 (C-3), 127.0 (C-5), 125.6 (C-7), 125.1 (C-4), 123.5 (C-2), 121.2 (C-15), 114.4 (C-14), 111.2 (C-11), 56.0 (C-16), 39.7 (C-18), 39.2 (C-9), 19.2 (C-19), 13.7 (C-20). HRMS (+ESI) [M + H]^+^: 326.1768, C_20_H_24_NO_3_ requires 326.1756.

*(E)-N-(2-(3-(4-Hydroxy-3-methoxyphenyl)prop-1-en-1-yl)phenyl)pentanamide* **8d**. IR (neat) ν: 3302 (w, OH), 3270 (w, NH), 3027 (w), 2961 (w), 1647 (s, C=O), 1510 (s), 1452 (m), 1377 (w), 1272 (m), 1229 (s), 1155 (m), 1039 (m, C−O), 850 (w), 818 (w), 782 (w), 755 (s) cm^−1^; ^1^H NMR (CDCl_3_, 500 MHz) δ: 7.80 (d, *J* = 7.5 Hz, H-2, 1H), 7.09–7.41 (m, NH, H-3, H-4, H-5, 4H), 6.72–6.73 (m, H-11, H-14, H-15, 3H), 6.43 (d, *J* = 15.3 Hz, H-7, 1H), 6.26 (dt, *J* = 15.3, 6.9 Hz, H-8, 1H), 5.64 (s, OH, 1H), 3.88 (s, H-16, 3H), 3.50 (d, *J* = 6.9 Hz, H-9, 2H), 2.30 (t, *J* = 7.5 Hz, H-18, 2H), 1.63–1.66 (m, H-19, 2H), 1.32–1.37 (m, H-20, 2H), 0.94 (t, *J* = 7.5 Hz, H-21, 3H); ^13^C NMR (CDCl_3_, 125.8 MHz) δ: 171.4 (C-17), 146.6 (C-13), 144.2 (C-12), 134.2 (C-1), 134.0 (C-8), 131.4 (C-10), 130.1 (C-6), 127.8 (C-3), 127.0 (C-5), 125.7 (C-7), 125.1 (C-4), 123.5 (C-2), 121.2 (C-15), 114.4 (C-14), 111.2 (C-11), 55.9 (C-16), 39.2 (C-9), 37.3 (C-18), 27.8 (C-19), 22.31 (C-20), 13.72 (C-21). HRMS (+ESI) [M + H]^+^: 340.1908, C_21_H_26_NO_3_ requires 340.1913.

*(E)-N-(2-(3-(4-Hydroxy-3-methoxyphenyl)prop-1-en-1-yl)phenyl)hexanamide* **8e**. IR (neat) ν: 3394 (w, OH), 3292 (w, NH), 3027 (w), 2955 (w), 1648 (s, C=O), 1511 (s), 1450 (m), 1372 (w), 1274 (s), 1236 (s), 1154 (m), 1038 (m, C−O), 851 (w), 817 (w), 785 (w), 754 (m) cm^−1^; ^1^H NMR (CDCl_3_, 500 MHz) δ: 7.81 (d, *J* = 8.1 Hz, H-2, 1H), 7.09–7.38 (m, NH, H-3, H-4, H-5, 4H), 6.72–6.73 (m, H-11, H-14, H-15, 3H), 6.44 (d, *J* = 15.6 Hz, H-7, 1H), 6.25 (dt, *J* = 15.6, 6.9 Hz, H-8, 1H), 5.60 (s, OH, 1H), 3.88 (s, H-16, 3H), 3.50 (d, *J* = 6.9 Hz, H-9, 2H), 2.30 (t, *J* = 7.2 Hz, H-18, 2H), 1.65–1.68 (m, H-19, 2H), 1.34–1.36 (m, H-20, H-21, 4H), 0.92 (t, *J* = 7.2 Hz, H-22, 3H); ^13^C NMR (CDCl_3_, 125.8 MHz) δ: 171.4 (C-17), 146.6 (C-13), 144.2 (C-12), 134.3 (C-1), 134.1 (C-8), 131.5 (C-10), 130.0 (C-6), 127.9 (C-3), 127.0 (C-5), 125.7 (C-7), 125.1 (C-4), 123.5 (C-2), 121.2 (C-15), 114.4 (C-14), 111.2 (C-11), 55.91 (C-16), 39.2 (C-9), 37.5 (C-18), 31.4 (C-20), 25.4 (C-19), 22.4 (C-21), 13.9 (C-22). HRMS (+ESI) [M + H]^+^: 354.2060, C_22_H_28_NO_3_ requires 354.2069.

*(E)-N-(2-(3-(4-Hydroxy-3-methoxyphenyl)prop-1-en-1-yl)phenyl)decanamide* **8f**. IR (neat) ν: 3399 (w, OH), 3271 (w, NH), 3068 (w), 2923 (m), 2853 (m), 1650 (s, C=O), 1510 (s), 1452 (m), 1377 (w), 1270 (m), 1226 (s), 1155 (m), 1039 (m, C−O), 850 (w), 822 (w), 787 (w), 753 (s) cm^−1^; ^1^H NMR (CDCl_3_, 500 MHz) δ: 7.83 (d, *J* = 7.9 Hz, H-2, 1H), 7.37 (d, *J* = 7.9 Hz, H-5, 1H), 7.32 (br s, NH, 1H), 7.30–7.32 (m, H-3, 1H), 7.07–7.09 (m, H-4, 1H), 6.73–6.74 (m, H-11, H-14, H-15, 3H), 6.44 (d, *J* = 15.4 Hz, H-7, 1H), 6.25 (dt, *J* = 15.4, 6.8 Hz, H-8, 1H), 5.54 (s, OH, 1H), 3.89 (s, H-16, 3H), 3.50 (d, *J* = 6.8 Hz, H-9, 2H), 2.30 (t, *J* = 7.3 Hz, H-18, 2H), 1.62–1.70 (m, H-19, 2H), 1.23–1.29 (m, H-20, H-21, H-22, H-23, H-24, H-25, 12H), 0.87 (t, *J* = 7.3 Hz, H-26, 3H); ^13^C NMR (CDCl_3_, 125.8 MHz) δ: 171.4 (C-17), 146.6 (C-13), 144.2 (C-12), 134.1 (C-1), 134.1 (C-8), 131.5 (C-10), 129.9 (C-6), 127.9 (C-3), 127.1 (C-5), 125.1 (C-7, C-4), 123.5 (C-2), 121.2 (C-15), 114.4 (C-14), 111.1 (C-11), 55.9 (C-16, OCH_3_), 39.3 (C-9), 37.7 (C-18), 31.8 (C-24), 29.5 (C-23), 29.4 (C-22), 29.34 (C-20), 29.27 (C-21), 25.8 (C-19), 22.6 (C-25), 14.1 (C-26). HRMS (+ESI) [M + H]^+^: 410.2691, C_26_H_36_NO_3_ requires 410.2695.

*(E)-N-(2-(3-(4-Hydroxy-3-methoxyphenyl)prop-1-en-1-yl)phenyl)cyclohexanecarboxamide* **8g**. IR (neat) ν: 3394 (w, OH), 3290 (m, NH), 3030 (w), 2928 (m), 1649 (s, C=O), 1512 (s), 1447 (m), 1386 (w), 1280 (m), 1226 (s), 1156 (m), 1039 (m, C−O), 859 (w), 830 (w), 783 (w), 750 (m) cm^−1^; ^1^H NMR (CDCl_3_, 500 MHz) δ: 7.84 (d, *J* = 8.0 Hz, H-2, 1H), 7.39 (br s, NH, 1H), 7.37 (t, *J* = 8.0 Hz, H-5, 1H), 7.25–7.29 (m, H-3, 1H), 7.08–7.11 (m, H-4, 1H), 6.73–6.74 (m, H-11, H-14, H-15, 3H), 6.38 (d, *J* = 15.3 Hz, H-7, 1H), 6.27 (dt, *J* = 15.3, 6.4 Hz, H-8, 1H), 5.57 (s, OH, 1H), 3.89 (s, H-16, 3H), 3.50 (d, *J* = 6.4 Hz, H-9, 2H), 2.15–2.21 (m, H-18, 1H), 1.92 (d, *J* = 11.5 Hz, H-19α, 2H), 1.75–1.77 (m, H-19β, 2H), 1.42–1.49 (m, H-20α, 2H), 1.16–1.28 (m, H-20β, H-21, 4H); ^13^C NMR (CDCl_3_, 125.8 MHz) δ: 174.2 (C-17), 146.6 (C-13), 144.2 (C-12), 134.3 (C-1), 134.2 (C-8), 131.3 (C-10), 129.4 (C-6), 127.8 (C-3), 127.0 (C-5), 125.6 (C-4), 125.0 (C-7), 123.4 (C-2), 121.3 (C-15), 114.4 (C-14), 111.3 (C-11), 55.92 (C-16), 46.3 (C-18), 39.2 (C-9), 29.8 (C-19), 25.7 (C-20, C-21). HRMS (+ESI) [M + H]^+^: 366.2065, C_23_H_28_NO_3_ requires 366.2069.

*(E)-N-(2-(3-(4-Hydroxy-3-methoxyphenyl)prop-1-en-1-yl)phenyl)benzamide* **8h**. IR (neat) ν: 3528 (w, OH), 3225 (w, NH), 3062 (w), 3027, 2965 (w), 1649 (m, C=O), 1516 (s), 1484 (m), 1372 (w), 1272 (s), 1247 (m), 1157 (m), 1026 (m, C−O), 866 (w), 819 (w), 793 (w), 754 (s) cm^−1^; ^1^H NMR (CDCl_3_, 500 MHz) δ: 6.68–8.22 (m, NH, H-3, H-4, H-5, H-11, H-14, H-15, H-19, H-20, H-21, 12H), 8.04 (d, *J* = 7.5 Hz, H-2, 1H), 6.47–6.50 (m, H-7, 1H), 6.30–6.37 (m, H-8, 1H), 5.50 (s, OH, 1H), 3.88 (s, H-16, 3H), 3.51 (d, *J* = 6.0 Hz, H-9, 2H); ^13^C NMR (CDCl_3_, 125.8 MHz), characteristic signals, δ: 134.7 (C-8), 134.3 (C-1), 128.8 (C-20), 127.9 (C-3), 125.0 (C-4), 123.2 (C-2), 121.3 (C-15), 114.3 (C-14), 111.2 (C-11), 55.85 (C-16), 39.2 (C-9). HRMS (+ESI) [M + H]^+^: 360.1596, C_23_H_22_NO_3_ requires 360.1600.

*(E)-N-(2-(3-(4-Hydroxy-3-methoxyphenyl)prop-1-en-1-yl)phenyl)-2-methylbenzamide* **8i**. IR (neat) ν: 3388 (w, OH), 3271 (w, NH), 3024 (w), 2965 (w), 1646 (s, C=O), 1515 (s), 1451 (m), 1379 (w), 1267 (s), 1236 (s), 1157 (w), 1032 (m, C−O), 866 (w), 834 (w), 790 (w), 751 (s) cm^−1^; ^1^H NMR (CDCl_3_, 500 MHz) δ: 8.00 (d, *J* = 8.0 Hz, H-2, 1H), 7.74 (br s, NH, 1H), 7.42 (d, *J* = 8.0 Hz, H-5, 1H), 7.36–7.40 (m, H-20, H-21, 2H), 7.35 (d, *J* = 7.5 Hz, H-23, 1H), 7.28 (t, *J* = 8.0 Hz, H-3, 1H), 7.15–7.21 (m, H-4, H-22, 2H), 6.68 (s, H-11, 1H), 6.72 (d, *J* = 8.2 Hz, H-14, H-15, 2H), 6.46 (d, *J* = 15.9 Hz, H-7, 1H), 6.30 (dt, *J* = 15.9, 6.3 Hz, H-8, 1H), 5.50 (s, OH, 1H), 3.80 (s, H-16, 3H), 3.48 (d, *J* = 6.3 Hz, H-9, 2H), 2.50 (s, H-24, 3H); ^13^C NMR (CDCl_3_, 125.8 MHz) δ: 167.9 (C-17), 146.3 (C-12), 144.3 (C-13), 136.7 (C-18), 136.4 (C-19), 134.3 (C-8), 134.1 (C-1), 131.3 (C-21), 131.2 (C-10), 130.24 (C-20), 130.0 (C-6), 127.9 (C-3), 127.1 (C-5), 126.5 (C-23), 126.1 (C-22), 125.8 (C-7), 125.4 (C-4), 125.0 (C-2), 121.2 (C-15), 114.37 (C-14), 111.2 (C-11), 55.7 (C-16), 39.1 (C-9), 19.9 (C-24). HRMS (+ESI) [M + H]^+^: 374.1743, C_24_H_24_NO_3_ requires 374.1756.

*(E)-N-(2-(3-(4-Hydroxy-3-methoxyphenyl)prop-1-en-1-yl)phenyl)furan-2-carboxamide* **8j**. IR (neat) ν: 3416 (w, OH), 3216 (w, NH), 3039 (w), 2954 (w), 1617 (m, C=O), 1511 (m), 1460 (m), 1350 (m), 1266 (m), 1227 (m), 1153 (m), 1035 (m, C−O), 885 (w), 826 (w), 785 (w), 758 (s) cm^−1^; ^1^H NMR (CDCl_3_, 500 MHz) δ: 8.43 (br s, NH, 1H), 8.12 (d, *J* = 7.9 Hz, H-21, 1H), 7.39 (d, *J* = 7.0 Hz, H-2, 1H), 7.29–7.33 (m, H-3, H-5, 2H), 7.15–7.20 (m, H-4, H-19, 2H), 6.78 (s, H-11, 1H), 6.75–6.77 (m, H-14, H-15, 2H), 6.55–6.58 (m, H-7, 1H), 6.54 (d, *J* = 7.9 Hz, H-20, 1H), 6.30–6.34 (m, H-8, 1H), 5.52 (s, OH, 1H), 3.88 (s, H-16, 3H), 3.54 (d, *J* = 6.0 Hz, H-9, 2H). HRMS (-ESI) [M-H]^−^: 348.1245, C_21_H_19_NO_4_ requires 348.1236.

*(E)-N-(2-(3-(4-Hydroxy-3-methoxyphenyl)allyl)phenyl)acetamide* **9a**. IR (neat) ν: 3348 (w, OH), 3229 (w, NH), 3125 (w), 2965 (w), 1636 (m, C=O), 1517 (s), 1460 (w), 1374 (w), 1273 (s), 1236 (m), 1175 (m), 1033 (m, C−O), 868 (w), 822 (w), 797 (w), 756 (s) cm^−1^; ^1^H NMR (CDCl_3_, 500 MHz) δ: 7.86 (d, *J* = 8.0 Hz, H-2′, 1H), 7.29 (br s, NH, 1H), 7.29 (ddd, *J* = 8.0, 7.5, 1.0 Hz, H-3′, 1H), 7.24 (d, *J* = 8.0 Hz, H-5′, 1H), 7.15 (dd, *J* = 8.0, 7.5 Hz, H-4′, 1H), 6.89−6.82 (m, H11′, H14′, H15′, 3H), 6.38 (dt, *J* = 16.0, 1.5 Hz, H-9′, 1H), 6.16 (dt, *J* = 16.0, 6.5 Hz, H-8′, 1H), 5.64 (s, OH, 1H), 3.89 (s, H-16′, 3H), 3.53 (dd, *J* = 6.5, 1.5 Hz, H-7′, 2H), 2.12 (s, H-18′, 3H); ^13^C NMR (CDCl_3_, 125 MHz) δ: 168.2 (C-17′), 146.7 (C-12′), 145.6 (C-13′), 136.1 (C-1′), 131.5 (C-9′), 130.4 (C-6′), 130.3 (C-5′), 129.2 (C-10′), 127.6 (C-3′), 125.4, 125.3 (C-4′, C-8′), 123.8 (C-2′), 120.0 (C-15′), 114.5 (C-14′), 108.0 (C-11′), 55.90 (C-16′), 36.1 (C-7′), 24.4 (C-18′). HRMS (+ESI) [M + H]^+^: 298.1435, C_18_H_20_NO_3_ requires 298.1443.

*(E)-N-(2-(3-(4-Hydroxy-3-methoxyphenyl)allyl)phenyl)isobutyramide* **9b**. IR (neat) ν: 3398 (w, OH), 3291 (m, NH), 3016 (w), 2970 (w), 1654 (s, C=O), 1509 (s), 1446 (m), 1371 (w), 1273 (s), 1236 (s), 1155 (m), 1038 (m, C−O), 848 (m), 812 (w), 798 (w), 753 (s) cm^−1^; ^1^H NMR (CDCl_3_, 500 MHz) δ: 7.95 (d, *J* = 8.0 Hz, H-2′, 1H), 7.35 (br s, NH, 1H), 7.29 (ddd, *J* = 8.0, 7.5, 1.0 Hz, H-3′, 1H), 7.23 (br d, *J* = 7.5 Hz, H-5′, 1H), 7.13 (t, *J* = 7.5 Hz, H-4′, 1H), 6.85 (s, H-11′, 1H), 6.84 (AB system, *δ*_A_ = 6.84, *δ*_B_ = 6.85, *J*_AB_ = 9.0 Hz, H15′, H14′, 2H), 6.38 (br d, *J* = 16.0 Hz, H-9′, 1H), 6.16 (dt, *J* = 16.0, 6.0 Hz, H-8′, 1H), 5.73 (br s, OH, 1H), 3.87 (s, H-16′, 3H), 3.53 (br d, *J* = 6.0 Hz, H-7′, 2H), 2.47 (sept, *J* = 7.0 Hz, H-18′, 1H), 1.21 (d, *J* = 7.0 Hz, H-19′, 6H); ^13^C NMR (CDCl_3_, 125.8 MHz) δ: 175.1 (C-17′), 146.6 (C-12′), 145.5 (C-13′), 136.3 (C-1′), 131.6 (C-9′), 130.2 (C-5′), 130.0 (C-6′), 129.2 (C-10′), 127.5 (C-3′), 125.1, 125.0 (C-4′, C-8′), 123.4 (C-2′), 119.9 (C-15′), 114.4 (C-14′), 108.0 (C-11′), 55.9 (C-16′), 36.7 (C-18′), 36.0 (C-7′), 19.6 (C-19′). HRMS (+ESI) [M + H]^+^: 326.1755, C_20_H_24_NO_3_ requires 326.1756.

*(E)-N-(2-(3-(4-Hydroxy-3-methoxyphenyl)allyl)phenyl)butyramide* **9c**. IR (neat) ν: 3399 (w, OH), 3269 (w, NH), 3053 (w), 2936 (w), 1646 (s, C=O), 1510 (s), 1453 (m), 1376 (w), 1271 (s), 1229 (s), 1155 (m), 1034 (m, C−O), 849 (w), 806 (w), 794 (w), 756 (s) cm^−1^; ^1^H NMR (CDCl_3_, 500 MHz) δ: 7.92 (d, *J* = 7.7 Hz, H-2′, 1H), 7.32 (br s, NH, 1H), 7.28 (t, *J* = 7.7 Hz, H-3′, 1H), 7.23 (d, *J* = 7.7 Hz, H-5′, 1H), 7.13 (t, *J* = 7.7 Hz, H-4′, 1H), 6.88 (s, H-11′, 1H), 6.84 (d, *J* = 8.2 Hz, H-15′, 1H), 6.82 (d, *J* = 8.2 Hz, H-14′, 1H), 6.38 (d, *J* = 15.9 Hz, H-9′, 1H), 6.16 (dt, *J* = 15.9, 6.2 Hz, H-8′, 1H), 5.69 (s, OH, 1H), 3.88 (s, H-16′, 3H), 3.52 (d, *J* = 6.2 Hz, H-7′, 2H), 2.29 (t, *J* = 7.3 Hz, H-18′, 2H), 1.71 (sext, *J* = 7.3 Hz, H-19′, 2H), 0.96 (t, *J* = 7.3 Hz, H-20′, 3H); ^13^C NMR (CDCl_3_, 125.8 MHz) δ: 171.2 (C-17′), 146.7 (C-12′), 145.5 (C-13′), 136.3 (C-1′), 131.6 (C-9′), 130.2 (C-5′), 130.0 (C-6′), 129.2 (C-10′), 127.5 (C-3′), 125.2, 125.1 (C-4′, C-8′), 123.5 (C-2′), 119.9 (C-15′), 114.4 (C-14′), 108.0 (C-11′), 55.9 (C-16′), 39.7 (C-18′), 36.1 (C-7′), 19.2 (C-19′), 13.7 (C-20′). HRMS (+ESI) [M + H]^+^: 326.1768, C_20_H_24_NO_3_ requires 326.1756.

*(E)-N-(2-(3-(4-Hydroxy-3-methoxyphenyl)allyl)phenyl)pentanamide* **9d**. IR (neat) ν: 3302 (w, OH), 3270 (w, NH), 3027 (w), 2961 (w), 1647 (s, C=O), 1510 (s), 1452 (m), 1377 (w), 1272 (m), 1229 (s), 1155 (m), 1039 (m, C−O), 850 (w), 818 (w), 782 (w), 755 (s) cm^−1^; ^1^H NMR (CDCl_3_, 500 MHz) δ: 7.91 (d, *J* = 7.8 Hz, H-2′, 1H), 7.37 (br s, NH, 1H), 7.28 (t, *J* = 7.8 Hz, H-3′, 1H), 7.23 (d, *J* = 7.8 Hz, H-5′, 1H), 7.13 (t, *J* = 7.8 Hz, H-4′, 1H), 6.86 (s, H-11′, 1H), 6.85 (d, *J* = 8.2 Hz, H-15′, 1H), 6.83 (d, *J* = 8.2 Hz, H-14′, 1H), 6.38 (d, *J* = 15.9 Hz, H-9′, 1H), 6.16 (dt, *J* = 15.9, 6.2 Hz, H-8′, 1H), 5.77 (s, OH, 1H), 3.87 (s, H-16′, 3H), 3.52 (d, *J* = 6.2 Hz, H-7′, 2H), 2.30 (t, *J* = 7.4 Hz, H-18′, 2H), 1.63–1.66 (m, H-19′, 2H), 1.32–1.37 (m, H-20′, 2H), 0.88 (t, *J* = 7.4 Hz, H-21′, 3H); ^13^C NMR (CDCl_3_, 125.8 MHz) δ: 171.4 (C-17′), 146.7 (C-12′), 145.5 (C-13′), 136.2 (C-1′), 131.6 (C-9′), 130.2 (C-5′), 130.1 (C-6′), 129.2 (C-10′), 127.5 (C-3′), 125.2, 125.1 (C-4′, C-8′), 123.5 (C-2′), 119.9 (C-15′), 114.5 (C-14′), 108.0 (C-11′), 55.8 (C-16′), 37.5 (C-18′), 36.1 (C-7′), 27.8 (C-19′), 22.34 (C-20′), 13.68 (C-21′). HRMS (+ESI) [M + H]^+^: 340.1908, C_21_H_26_NO_3_ requires 340.1913.

*(E)-N-(2-(3-(4-Hydroxy-3-methoxyphenyl)allyl)phenyl)hexanamide* **9e**. IR (neat) ν: 3394 (w, OH), 3292 (w, NH), 3027 (w), 2955 (w), 1648 (s, C=O), 1511 (s), 1450 (m), 1372 (w), 1274 (s), 1236 (s), 1154 (m), 1038 (m, C−O), 851 (w), 817 (w), 785 (w), 754 (m) cm^−1^; ^1^H NMR (CDCl_3_, 500 MHz) δ: 7.92 (d, *J* = 7.7 Hz, H-2′, 1H), 7.36 (br s, NH, 1H), 7.28 (t, *J* = 7.7 Hz, H-3′, 1H), 7.24 (d, *J* = 7.7 Hz, H-5′, 1H), 7.13 (t, *J* = 7.7 Hz, H-4′, 1H), 6.86 (s, H-11′, 1H), 6.85 (d, *J* = 8.2 Hz, H-15′, 1H), 6.84 (d, *J* = 8.2 Hz, H-14′, 1H), 6.38 (d, *J* = 15.9 Hz, H-9′, 1H), 6.15 (dt, *J* = 15.9, 6.2 Hz, H-8′, 1H), 5.73 (s, OH, 1H), 3.87 (s, H-16′, 3H), 3.52 (d, *J* = 6.2 Hz, H-7′, 2H), 2.30 (t, *J* = 7.5 Hz, H-18′, 2H), 1.65–1.68 (m, H-19′, 2H), 1.28–1.29 (m, H-20′, H-21′, 4H), 0.86 (t, *J* = 7.5 Hz, H-22′, 3H); ^13^C NMR (CDCl_3_, 125.8 MHz) δ: 171.4 (C-17′), 146.7 (C-12′), 145.6 (C-13′), 136.3 (C-1′), 131.6 (C-9′), 130.2 (C-5′), 130.0 (C-6′), 129.2 (C-10′), 127.5 (C-3′), 125.2, 125.1 (C-4′, C-8′), 123.5 (C-2′), 119.9 (C-15′), 114.5 (C-14′), 108.0 (C-11′), 55.86 (C-16′), 37.8 (C-18′), 36.1 (C-7′), 31.4 (C-20′), 25.4 (C-19′), 22.3 (C-21′), 13.8 (C-22′). HRMS (+ESI) [M + H]^+^: 354.2060, C_22_H_28_NO_3_ requires 354.2069.

*(E)-N-(2-(3-(4-Hydroxy-3-methoxyphenyl)allyl)phenyl)decanamide* **9f**. IR (neat) ν: 3399 (w, OH), 3271 (w, NH), 3068 (w), 2923 (m), 2853 (m), 1650 (s, C=O), 1510 (s), 1452 (m), 1377 (w), 1270 (m), 1226 (s), 1155 (m), 1039 (m, C−O), 850 (w), 822 (w), 787 (w), 753 (s) cm^−1^; ^1^H NMR (CDCl_3_, 500 MHz) δ: 7.93 (d, *J* = 6.8 Hz, H-2′, 1H), 7.32 (br s, NH, 1H), 7.29 (t, *J* = 6.8 Hz, H-3′, 1H), 7.23 (d, *J* = 6.8 Hz, H-5′, 1H), 7.13 (t, *J* = 6.8 Hz, H-4′, 1H), 6.86 (s, H-11′, 1H), 6.85 (d, *J* = 8.2 Hz, H-15′, 1H), 6.83 (d, *J* = 8.2 Hz, H-14′, 1H), 6.38 (d, *J* = 15.9 Hz, H-9′, 1H), 6.16 (dt, *J* = 15.9, 6.2 Hz, H-8′, 1H), 5.65 (s, OH, 1H), 3.88 (s, H-16′, 3H), 3.52 (d, *J* = 6.2 Hz, H-7′, 2H), 2.30 (t, *J* = 7.3 Hz, H-18′, 2H), 1.62–1.70 (m, H-19′, 2H), 1.23–1.29 (m, H-20′, H-21′, H-22′, H-23′, H-24′, H-25′, 12H), 0.87 (t, *J* = 7.3 Hz, H-26′, 3H); ^13^C NMR (CDCl_3_, 125.8 MHz) δ: 171.4 (C-17′), 146.7 (C-12′), 145.5 (C-13′), 136.3 (C-1′), 131.6 (C-9′), 130.2 (C-5′), 129.9 (C-6′), 129.2 (C-10′), 127.6 (C-3′), 125.2 (C-4′, C-8′), 123.45 (C-2′), 120.0 (C-15′), 114.5 (C-14′), 107.9 (C-11′), 55.87 (C-16′), 37.9 (C-18′), 36.2 (C-7′), 31.8 (C-24′), 29.4 (C-23′), 29.32 (C-22′), 29.29 (C-20′), 29.2 (C-21′), 25.8 (C-19′), 22.6 (C-25′), 14.1 (C-26′). HRMS (+ESI) [M + H]^+^: 410.2691, C_26_H_36_NO_3_ requires 410.2695.

*(E)-N-(2-(3-(4-Hydroxy-3-methoxyphenyl)allyl)phenyl)cyclohexanecarboxamide* **9g**. IR (neat) ν: 3394 (w, OH), 3290 (m, NH), 3030 (w), 2928 (m), 1649 (s, C=O), 1512 (s), 1447 (m), 1386 (w), 1280 (m), 1226 (s), 1156 (m), 1039 (m, C−O), 859 (w), 830 (w), 783 (w), 750 (m) cm^−1^; ^1^H NMR (CDCl_3_, 500 MHz) δ: 7.96 (d, *J* = 7.6 Hz, H-2′, 1H), 7.39 (br s, NH, 1H), 7.28 (t, *J* = 7.6 Hz, H-3′, 1H), 7.23 (d, *J* = 7.6 Hz, H-5′, 1H), 7.11 (t, *J* = 7.6 Hz, H-4′, 1H), 6.87 (d, *J* = 8.2 Hz, H-15′, 1H), 6.85 (s, H-11′, 1H), 6.84 (d, *J* = 8.2 Hz, H-14′, 1H), 6.41 (d, *J* = 15.9 Hz, H-9′, 1H), 6.16 (dt, *J* = 15.9, 6.2 Hz, H-8′, 1H), 5.69 (s, OH, 1H), 3.88 (s, H-16′, 3H), 3.53 (d, *J* = 6.2 Hz, H-7′, 2H), 2.15–2.21 (m, H-18′, 1H), 1.92 (d, *J* = 11.5 Hz, H-19α′, 2H), 1.75–1.77 (m, H-19β′, 2H), 1.42–1.49 (m, H-20α′, 2H), 1.16–1.28 (m, H-20β′, H-21′, 4H); ^13^C NMR (CDCl_3_, 125.8 MHz) δ: 174.2 (C-17′), 146.7 (C-12′), 145.5 (C-13′), 136.4 (C-1′), 131.8 (C-9′), 130.2 (C-5′), 129.8 (C-6′), 129.2 (C-10′), 127.5 (C-3′), 125.2 (C-8′), 124.9 (C-4′), 123.3 (C-2′), 120.0 (C-15′), 114.5 (C-14′), 107.9 (C-11′), 55.87 (C-16′), 46.5 (C-18′), 36.2 (C-7′), 29.8 (C-19′,), 25.7 (C-20′, C-21′). HRMS (+ESI) [M + H]^+^: 366.2065, C_23_H_28_NO_3_ requires 366.2069.

*(E)-N-(2-(3-(4-Hydroxy-3-methoxyphenyl)allyl)phenyl)benzamide* **9h**. IR (neat) ν: 3528 (w, OH), 3225 (w, NH), 3062 (w), 3027, 2965 (w), 1649 (m, C=O), 1516 (s), 1484 (m), 1372 (w), 1272 (s), 1247 (m), 1157 (m), 1026 (m, C−O), 866 (w), 819 (w), 793 (w), 754 (s) cm^−1^; ^1^H NMR (CDCl_3_, 500 MHz) δ: 8.22 (s, NH, 1H), 8.12 (d, *J* = 7.5 Hz, H-2′, 1H), 7.77 (d, *J* = 7.5 Hz, H-19′, 2H), 7.51 (t, *J* = 7.5 Hz, H-21′, 1H), 7.36 (t, *J* = 7.5 Hz, H-3′, H-20′, 3H), 7.29 (d, *J* = 7.5 Hz, H-5′, 1H), 7.18 (t, *J* = 7.5 Hz, H-4′, 1H), 6.87 (s, H-11′, 1H), 6.86 (AB system, *δ*_A_ = 6.84, *δ*_B_ = 6.88, *J*_AB_ = 8.5 Hz, H15′, H14′, 2H), 6.46 (br d, *J* = 16.0 Hz, H-9′, 1H), 6.22 (dt, *J* = 16.0, 5.5 Hz, H-8′, 1H), 5.66 (s, OH, 1H), 3.88 (s, H-16′, 3H), 3.61 (d, *J* = 5.5 Hz, H-7′, 2H); ^13^C NMR (CDCl_3_, 125.8 MHz) δ: 165.4 (C-17′), 146.7 (C-12′), 145.6 (C-13′), 136.7 (C-1′), 134.8 (C-18′), 132.2 (C-9′), 131.8 (C-21′), 130.4 (C-5′), 129.9 (C-6′), 128.9 (C-10′), 128.8 (C-20′), 127.8 (C-3′), 127.1 (C-19′), 125.2 (C-8′), 125.0 (C-4′), 123.2 (C-2′), 120.2 (C-15′), 114.5 (C-14′), 107.9 (C-11′), 55.87 (C-16′), 36.8 (C-7′). HRMS (+ESI) [M + H]^+^: 360.1596, C_23_H_22_NO_3_ requires 360.1600.

*(E)-N-(2-(3-(4-Hydroxy-3-methoxyphenyl)allyl)phenyl)-2-methylbenzamide* **9i**. IR (neat) ν: 3388 (w, OH), 3271 (w, NH), 3024 (w), 2965 (w), 1646 (s, C=O), 1515 (s), 1451 (m), 1379 (w), 1267 (s), 1236 (s), 1157 (w), 1032 (m, C−O), 866 (w), 834 (w), 790 (w), 751 (s) cm^−1^; ^1^H NMR (CDCl_3_, 500 MHz) δ: 8.16 (d, *J* = 7.9 Hz, H-2′, 1H), 7.74 (br s, NH, 1H), 7.36–7.40 (m, H-20′, H-21′, 2H), 7.32 (d, *J* = 7.5 Hz, H-23′, 1H), 7.28 (t, *J* = 7.9 Hz, H-3′, 1H), 7.23 (d, *J* = 7.9 Hz, H-5′, 1H), 7.17 (t, *J* = 7.5 Hz, H-22′, 1H), 7.09 (t, *J* = 7.9 Hz, H-4′, 1H), 6.76 (s, H-11′, 1H), 6.83 (d, *J* = 8.2 Hz, H-14′, H-15′, 2H), 6.28 (d, *J* = 15.9 Hz, H-9′, 1H), 6.14 (dt, *J* = 15.9, 6.2 Hz, H-8′, 1H), 5.61 (s, OH, 1H), 3.86 (s, H-16′, 3H), 3.56 (d, *J* = 6.2 Hz, H-7′, 2H), 2.47 (s, H-24′, 3H); ^13^C NMR (CDCl_3_, 125.8 MHz) δ: 167.9 (C-17′), 146.6 (C-12′), 145.5 (C-13′), 136.7 (C-18′), 136.5 (C-19′), 136.2 (C-1′), 131.8 (C-9′), 131.3 (C-21′), 130.3 (C-5′), 130.15 (C-20′), 130.0 (C-6′), 129.0 (C-10′), 127.6 (C-3′), 126.7 (C-23′), 125.9 (C-22′), 125.5 (C-4′), 125.2 (C-8′), 124.8 (C-2′), 119.9 (C-15′), 114.39 (C-14′), 108.0 (C-11′), 55.8 (C-16′), 36.3 (C-7′), 19.9 (C-24′). HRMS (+ESI) [M + H]^+^: 374.1743, C_24_H_24_NO_3_ requires 374.1756.

*(E)-N-(2-(3-(4-Hydroxy-3-methoxyphenyl)allyl)phenyl)furan-2-carboxamide* **9j**. IR (neat) ν: 3416 (w, OH), 3216 (w, NH), 3039 (w), 2954 (w), 1617 (m, C=O), 1511 (m), 1460 (m), 1350 (m), 1266 (m), 1227 (m), 1153 (m), 1035 (m, C−O), 885 (w), 826 (w), 785 (w), 758 (s) cm^−1^; ^1^H NMR (CDCl_3_, 500 MHz) δ: 8.43 (br s, NH, 1H), 8.12 (d, *J* = 7.9 Hz, H-21′, 1H), 7.29–7.33 (m, H-2′, H-3′, H-5′, 3H), 7.15–7.20 (m, H-4′, H-19′, 2H), 6.89 (s, H-11′, 1H), 6.87 (d, *J* = 8.2 Hz, H-15′, 1H), 6.86 (d, *J* = 8.2 Hz, H-14′, 1H), 6.55 (d, *J* = 15.9 Hz, H-9′, 1H), 6.54 (d, *J* = 7.9 Hz, H-20′, 1H), 6.18 (dt, *J* = 15.9, 6.2 Hz, H-8′, 1H), 5.62 (s, OH, 1H), 3.88 (s, H-16′, 3H), 3.62 (d, *J* = 6.2 Hz, H-7′, 2H); ^13^C NMR (CDCl_3_, 125.8 MHz) δ: 156.1 (C-17′), 148.0 (C-18′), 146.7 (C-12′), 145.5 (C-13′), 144.1 (C-21′), 136.0 (C-1′), 132.3 (C-9′), 130.3 (C-5′), 130.0 (C-6′), 129.4 (C-10′), 127.7 (C-3′), 125.1 (C-4′), 124.7 (C-8′), 122.9 (C-2′), 120.2 (C-15′), 115.1 (C-19′), 114.3 (C-14′), 112.5 (C-20′), 107.7 (C-11′), 55.9 (C-16′), 36.6 (C-7′). HRMS (-ESI) [M-H]^−^: 348.1245, C_21_H_19_NO_4_ requires 348.1236.

### 3.3. Bioassay Assessment

#### 3.3.1. Cell Cultures

MCF-7 and MDA-MB-231 human breast cancer cell lines, as well as MCF-10A human normal breast cell lines, were used in this study. These cells were acquired from the American Type Culture Collection (Manassas, VA, USA). MCF-7 and MDAMB-231 cells were regularly cultured in Dulbecco’s Modified Eagle’s Medium (DMEM; Gibco, Waltham, MA, USA), and supplemented with 10% (*v*/*v*) foetal bovine serum (FBS; Gibco, Waltham, MA, USA) and 100 U/mL penicillin-streptomycin (PenStrep; Gibco, Waltham, MA, USA). MCF-10A cells were routinely cultured in DMEM/F12 (Gibco, Waltham, MA, USA) medium supplemented with 5% horse serum (Sigma-Aldrich, Burlington, MA, USA), 10 μg/mL of insulin (Sigma-Aldrich, Burlington, MA, USA), 20 ng/mL of human epidermal growth factor (hEGF; ThermoFisher, Waltham, MA, USA), 0.5 μg/mL of hydrocortisone (Sigma-Aldrich, Burlington, MA, USA) and 1 U/mL PenStrep. All cell lines were maintained in a humidified atmosphere with 5% CO_2_ at 37 °C.

#### 3.3.2. Semi-Empirical Calculations

Molecular modelling was performed on compounds **6a**, **7a**, **8a**, **9a**, **6b**, **7b**, **8b** and **9b**. In each case, the most stable conformer was located and its geometry pre-optimised using the molecular mechanics MMFF94 force field [38], with the Avogadro 1.2.0 software [39,40]. Semi-empirical geometry optimisation was then performed using the PM7 method [41], with the MOPAC2016 software [42].

#### 3.3.3. Cytotoxicity Assays

The effects of the *N*-(2-cinnamylphenyl)amide products (i.e., isolated pure isomers) on MCF-7, MDA-MB-231 and MCF-10A were determined using a 3-(4,5-dimethylthiazol-2-yl)-2,5-diphenyltetrazolium bromide (MTT) proliferation assay. The cell lines were seeded in 96-well plates at a density of 1 × 10^4^ cells/well and allowed to grow overnight for cell attachment. The cell lines were then treated with fresh assay medium, supplemented with increasing concentrations of tested compounds (10–100 µM) and incubated for 24–72 h at 37 °C. For this assay, the blank and negative controls used were the culture medium alone (without cells) and cells with culture medium only (untreated cells), respectively. For the positive control, MCF-10A cells were treated with 30 μM tamoxifen. At each incubation period (24, 48 and 72 h), 10 μL of MTT solution (5 mg/mL) was added to each well, and cells were further incubated for 4 h at 37 °C in 5% CO_2_. The solution mixture in each well was replaced with 100 μL of dimethyl sulfoxide (DMSO) to solubilise the MTT crystalline products. The optical density (O.D.) of each well was measured using a microplate reader (PowerWave-XS; Bio-Tek Instruments, Winooski, Vermont, USA) at 570 nm. The experiments were conducted in triplicate for each cell line.

A plot of % cell proliferation versus sample concentration was used to show the 50% inhibitory concentration (IC_50_). The selectivity index (SI) was used to indicate the cytotoxic selectivity of the samples against cancer cells and normal cells. It was calculated from the IC_50_ of the samples in normal cells versus cancer cells. An SI of less than 2 suggests the general toxicity of the drug/sample in cells, while an SI of more than 2 is an indication of the sample selectively targeting cancer cells [37]. All data were analysed and presented as means standard deviation (±SD). Comparisons between each concentration of 18 samples were evaluated using Student’s *t*-test, where *p* < 0.05, *p* < 0.01 or *p* < 0.001 were considered as statistically significant relative to the untreated control cells.

## 4. Conclusions

A series of new bio-inspired (*E*)-1,3-diarylpropene derivatives were synthesised using palladium-catalysed Heck cross-coupling reactions of 2-amidoiodobenzene derivatives with either estragole or eugenol, in 36–91% yields. In each case, two regioisomers were obtained, differing by the position of the (*E*) C=C double bond in the central propene subunit. Higher selectivities were obtained with eugenol, which was rationalised by the β-hydride elimination step being kinetically favoured at the position adjacent to the most electron-rich aromatic substituent. The products could be considered as stilbenoid and dihydrostilbenoid structural analogues, as well as hybrid molecules, and their cytotoxicities were evaluated using MTT proliferation assays against several cancer cell lines. However, only compounds **6d**, **6e**, **7d**, **7i** and **7e** exhibited low growth inhibition against MCF-7 and MDA-MB-231, respectively. Hence, no promising effects were disclosed, in comparison with the reference drug tamoxifen.

## Data Availability

Not applicable.

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
