# Peer review of "Synthesis of Bio-Inspired 1,3-Diarylpropene Derivatives via Heck Cross-Coupling and Cytotoxic Evaluation on Breast Cancer Cells"

_molecules, 2022, doi:10.3390/molecules27175373_

Round 1

Reviewer 1 Report

The manuscript authored by Azmi et al. reports on synthesis of various 1,3-diarylpropnenes via the Heck reaction and their bioactivity against several cancer cell lines. The authors describe in detail synthetic procedures, compound characterization (NMR), and mechanistic considerations that explain the observed regioselectivity in the final products. The details are in many cases trivial, redundant, and inconsequential for the overall content of the manuscript. The essential part of it is also cytotoxic testing of the prepared compounds. However, none of them had an activity that would be even close to that of tamoxifen.

Comments:

a) The synthesized compounds, 1,3-diarylpropenes, are not analogues of stilbeneiods or dihydrostilbenoids (page 2, the last paragraph). This claim is not correct; hence they cannot be envisioned to possess similar properties just because they bear similarly substituted arene rings.

b) The claim regarding the observed regioselectivity and energetics of the transition states might be correct (page 7). On the other hand, the authors did not provide any experimental or computational evidence that would support such a suggestion. This aspect should be clarified and appropriate measures taken to support such a claim.

Author Response

Dear sir

Thank you for the comments. Please refer the attachment. 

Reviewer 2 Report

This manuscript, submitted by Azmi and co-workers, describes synthesis of various 1,3-diarylpropenes and evaluation of cytotoxic activity against breast cancer cells such as MCF-7. In the synthesis, Heck reaction was employed to connect two aromatic rings. Beta-elimination step, an important process in Heck reaction, induced mixture of two isomers which is thought to be this reaction complicated. To address this process, calculation and consideration were mentioned in this manuscript. Although cytotoxic evaluation with the synthetic sample did not show a remarkable activity, compound 7d exhibits moderate against MCF-7. Taking these results in consideration, this referee thinks this manuscript should be accepted after addressing the following points.

1. Although it is mentioned that pure 9b was obtained, were other products separated to be investigated the cytotoxicities? If you separated compound, please mention in text or supplementary. If not, please mention a mixture was subjected to the test.

2. How did authors characterize 1H and 13C NMR although it is mentioned that only 9b was obtained as a pure compound? Please show how the mixture was characterized.

3. About 13C NMR assignments, number of peaks of product is missing in all cases I checked. Check again carefully.

4. In scheme 2, please check nC5H11.

Author Response

Dear sir

Thank you for the comments. Please find enclosed the attachment for the answers to your comments. 

Reviewer 3 Report

This study used the Heck reaction to synthesise a series of stilbene analogues for the purpose of testing their potency for killing several human breast cancer cell lines in culture. The structural class was chosen becaasue of publications about the antioxidant potential of the stilbene resveratrol and medicinal chemistry studies of analogues (eg, Molecules. 2020 Feb; 25(4): 893).

The work was clearly described with careful attention given to describing the mix of isomers expected from the reaction. None were sufficiently active over the 3-day exposure period to warrant use against cancer. Longer exposure time, say 3-5 doublings, would have likely given increased potency but not asltered the conclusion. 

The work would have more impact if the compounds had been tested against a wider range of cell lines, as reported in the above paper; or if they had been assayed for some of tthe other properties noted for resveratrol. 

Author Response

(The authors gave the same response as above.)

Round 2

Reviewer 1 Report

The authors have adequately replied to the raised issues.

Author Response

Dear reviewer
Thank you for the comment and suggestions. 

Reviewer 2 Report

This revised manuscript submitted by Six and Azmi describes syntheses of various 1,3-diarylpropenes and evaluations of cytotoxicity with the synthetic samples towards breast cancer cells. I am a reviewer 2. The comments I made before have been mentioned or improved in the revised manuscript . Thus,  I believed this should be accepted in Molecules.

Author Response

(The authors gave the same response as above.)
